# INPUT TIME SCALING: FINDINGS WITH SMALL&LOW QUALITY DATASETS, NOISE AND EFFICIENCY

## ABSTRACT

Large Language Models (LLMs) excel at mathematical reasoning, traditionally requiring high-quality large-scale data and extensive training. Recent works reveal a **Less-Is-More** phenomenon where very small, carefully curated high-quality datasets match resource-intensive approaches. In this work, we further systematically relax quality constraints under the Less-Is-More datasets by adding controlled noise and using different quality datasets. We control noise levels via persona context relevance to original queries. Counterintuitively, mixing relevant and irrelevant personas during different stages yields optimal results, and performance gains emerge only when context concatenation applies consistently, not necessarily the same type, across training and inference. Token distribution analysis shows that applying persona strategies increases thinking tokens while reducing response length, making reasoning more efficient. We term the above phenomenon **training-testing co-design**. Comparing dataset qualities, high-quality data excels on weaker models and easier questions, while low-quality data achieves overall higher scores, especially on hard questions with sufficiently capable models. Building on these insights, we propose our method, applying small and low-quality data to capable enough models with training-testing co-design. The process distinguishes itself from supervised fine-tuning or test-time scaling, which we term **Input-Time Scaling**. Our method achieves 76.7% pass@1 on AIME24/AIME25 using Qwen2.5-32B-Instruct, with DeepSeek-R1-Distill-Qwen-32B reaching 90.0%/80.0%, which gets state-of-the-art performance among Qwen2.5-32B variants. We are open-sourcing our datasets, pipelines, evaluation results, and checkpoints to facilitate reproducibility and further research.

## 1 INTRODUCTION

Modern large language models (LLMs) exhibit exceptional mathematical reasoning capabilities [Guha et al. (2025); Li et al. (2025c); Guo et al. (2025); He et al. (2025)], traditionally attributed to meticulously curated large-scale datasets and sophisticated training pipelines. Current approaches typically employ a two-stage paradigm: extensive supervised fine-tuning followed by reinforcement learning optimization to further boost performance. However, this conventional wisdom presents significant practical barriers [Havrilla et al. (2024); Zhang et al. (2024); Ye et al. (2025); Li et al. (2025a)]. Dataset curation demands substantial human expertise, quality filtering relies on nontrivial inductive biases, and computational requirements strain community resources. Recent **Less-is-More** adaptations [Ye et al. (2025); Muennighoff et al. (2025)] demonstrate that very small, carefully curated high-quality datasets can achieve competitive performance, eliminating the requirement for large dataset sizes. And in this way, the computational resources needed are also much smaller. However, these approaches still require substantial human labor to curate such datasets. From their latent intuition on high-quality data, a fundamental tension emerges: **can we further relax the requirement of data quality, with almost no human labor?**

To explore this fundamental problem more deeply, we relax data quality constraints by examining different dimensions. The common heuristic of quality can be abstracted as **"garbage in, garbage out"**, where low-quality information should be removed from query-answer pairs. Most human labor in curating high-quality data is spent on this task. To check the necessity of this intuition,

we sought to find a method that can automatically add different levels of noise into query-answer pairs. By controlling the noise levels, we compare directly their impacts on performances. However, naively adding noise or rewriting the data will provide uncontrollable issues, which will require hard work on further quality assessment. Inspired by current meta-cognition methods [Kaur et al. (2024); Didolkar et al. (2024); Wang et al. (2025b)] where LLMs demonstrate the ability to generate required persona contexts given information, we employ meta-cognition methods to automatically generate such contexts. We use the relevance between the personas and the original queries as a proxy to noisy levels. We then concatenate the persona contexts to queries, without modifying the original query and answer pair. By controlling the relevance between added personas and queries, we control the data quality with the corresponding noise level heuristically. To further ablate internal data quality between different datasets, we build our experiments comparing the carefully curated dataset with a considerably low-quality dataset that has only query filtering but no answer (CoT) filtering. If the data quality heuristics hold, adding relevant personas should outperform adding irrelevant ones, and high-quality datasets should boost the performance better than low-quality ones clearly.

Through extensive experiments, however, we discover counterintuitive findings. Adding noise does not necessarily degrade performance and can even improve it. During investigation of token distributions, applying persona strategies makes the reasoning process more efficient, thus improving performance. For datasets of different quality levels, low-quality datasets excel when the model is weak and focuses on easy questions. As the model becomes increasingly capable, low-quality datasets can actually boost overall performance, especially on hard questions. Overall, the quality requirements do not hold in our experiments. From our analysis, we discover the **training-testing co-design** phenomenon: adding any persona context during both training and testing, not necessarily the same type, boosts performance. Thus, we frame our method as applying small, low-quality datasets to sufficiently capable models with persona strategies during both training and testing. Our experiments validate these conclusions across a broad range of models and parameter sizes, from Qwen2.5 to Llama3 series. We name the overall empirical design **Input-Time Scaling**, and we achieve state-of-the-art performance on 32B Qwen2.5 model variants. With only 1K training examples, we achieve 76.7% pass@1 on both AIME24 and AIME25. Starting from DeepSeek-R1-Distill-Qwen-32B, we achieve 90% and 80% pass@1 on AIME24 and AIME25, respectively. Notably, this performance is achieved without further reinforcement learning, which may serve as a new starting point for further improvements. **Our contributions** are listed here:

1. We systematically relax data quality constraints by adding controlled noise and using different quality datasets. Counterintuitively, mixing relevant and irrelevant contexts yields optimal results, while low-quality data can achieve much higher scores.

2. We observe the training-testing co-design phenomenon. Performance gains emerge only when context concatenation applies consistently, though not necessarily of the same type, across training and inference. This increases thinking tokens while reducing response length, making reasoning more efficient.

3. We recognize the different learning patterns of low-quality and high-quality datasets. Low-quality data especially benefits hard questions with capable models, while high-quality data excels on weaker models and easier questions.

4. We propose Input-Time Scaling and achieve state-of-the-art performance. It is an empirical paradigm built on previous findings, applying small (1K), low-quality data to capable models via training-testing co-design with minimal human labor. Training on Qwen2.5-32B-Instruct achieves 76.7% pass@1 on both AIME24 and AIME25, and DeepSeek-R1-Distill-Qwen-32B achieves 90% (AIME24) and 80% (AIME25). This performance is comparable to models over 10× larger trained with 100× more data.

To facilitate reproducibility and further research, we are open-sourcing our datasets, data pipelines, evaluation results, and checkpoints. Our pipeline is extremely simple and clear, enabling straightforward reproduction of the reported results.

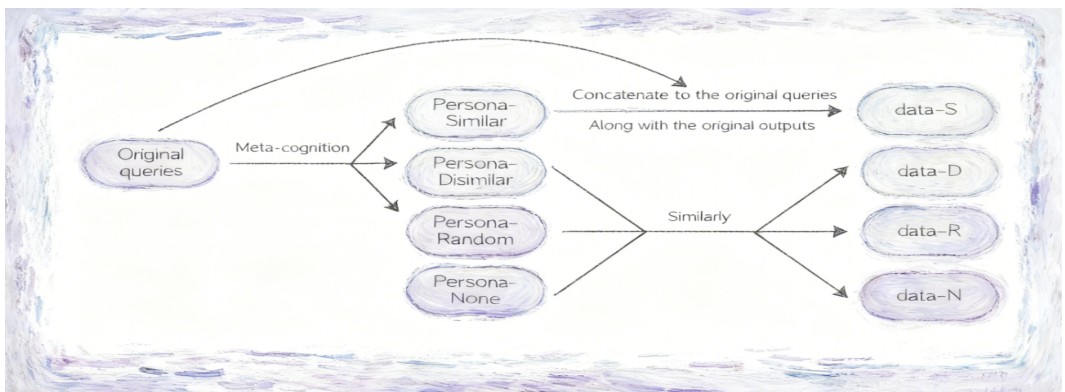

Figure 1: Data processing pipeline overview. We use meta-cognition methods to create three kinds of persona contexts concatenated to queries. For each persona type, we generate corresponding persona contexts from the original queries, then concatenate these contexts to the queries. The concatenated queries plus the original cot&answers form the corresponding dataset. Note that each query receives a unique persona context with no reuse. Overall, we create four datasets: three with different persona types and one with the original unchanged dataset. By controlling the relevance between personas and queries, we control the noise levels added to the queries.

## 2 METHODS: SYSTEMATICALLY EXPLORING THE QUALITY REQUIREMENT OF LESS-IS-MORE METHODS

While recent **Less-is-More** adaptations demonstrate that small, high-quality datasets can achieve competitive performance, these approaches still require substantial human labor to curate such datasets. **Can we further relax the data quality requirements with almost no human labor?** To explore this fundamental question, we examine different dimensions of data quality constraints: **(1)** What happens if we add noisy information to query-answer data pairs? **(2)** Do higher-quality datasets show better results?

### 2.1 ADDING IRRELEVANT/IRRELEVANT CONTEXT INTO QUERY-ANSWER PAIRS

It is common sense to use **garbage in, garbage out** as the guiding heuristic. Under this principle, removing low-quality information from query-answer pairs requires the most human labor. Instead, we focus on automatically adding noise to the query-answer pairs to examine whether the noise indeed degrades performance. Directly modifying the query&answer pair may degrade the overall quality; more importantly, comparing the quality of modifications is non-trivial. Inspired by current meta-cognition methods, LLMs can generate requested persona contexts based on intentions. We use the relevance between the persona context and queries as a proxy for the noise level we add. We then concatenate the generated personas to the original queries. In this way, we keep the original query and answer unchanged, while we control the noise level added intuitively. Specifically, we prompt DeepSeek-R1-0528 to generate personas, keeping all the default parameters. We prompt to generate personas with three categories: relevant, irrelevant, and random. Along with these three types, concatenating nothing is another type to contrast the three strategies using personas. We name them as follows:

- **Persona-None (N)**: Concatenate nothing to the query.

- **Persona-Similar (S)**: Generate a persona related to the query. Concatenate it to the query.

- **Persona-Dissimilar (D)**: Generate a persona not related to the query that cannot provide any useful information for the query. Concatenate it to the query.

- **Persona-Random (R)**: Randomly choose a domain and generate a corresponding persona. Concatenate it to the query. (Domains are generated by pure meta-cognition of human knowledge domains, which you can find details in the appendix A.6)

You can find detailed prompts and examples in AppendixA.6, and the visualized process in Figure1. The generated personas are intuitively relevant or irrelevant to the provided query. Adding context related to the query can provide background that is supposed to help the model better understand the problem, thus boosting performance. On the other hand, adding irrelevant context will distract the model, which can be understood as noise. For random personas, the domain may or may not relate to mathematics; thus, they should behave like a mixture of the two methods. We test the assumptions in later experiments.

## 2.2 COMPARING THE EFFECTS BETWEEN HIGH AND LOW QUALITY DATASETS

We adapt LIMO and OpenThought as two dataset sources. LIMO[Ye et al. (2025)] is a high-quality dataset that contains fewer than 1k problem-answer pairs. It is curated through careful query selection and, more importantly, through dedicated reasoning chain construction and answer filtering. OpenThought[Guha et al. (2025)] is a lower-quality dataset that contains more than 1M problem-answer pairs, without any filtering on CoT and answers. Many data points have no answer at all. Furthermore, each query in the OpenThought dataset has been sampled 16 times using QwQ-32B. If we randomly select examples from this dataset, the query diversity is also inferior compared to LIMO. As a result, we regard LIMO as the higher-quality dataset and OpenThought as the lower-quality one.

## 3 EXPERIMENTS

We primarily compare our results to LIMO[Ye et al. (2025)], S1[Muennighoff et al. (2025)], OpenThought[Guha et al. (2025)], MiroMind-M1[Li et al. (2025c)], and Skywork-OR1[He et al. (2025)]; consequently, we focus mainly on the 32B model size. We select the Qwen2.5[Team (2024)] series as they represent the most widely used non-reasoning models and are common base models in comparisons. This allows us to analyze the performance gains purely from our training-testing designs. We further demonstrate the effectiveness of our methods on the DeepSeek-R1-Distill-Qwen[Guo et al. (2025)] series to show that our methods can also benefit reasoning models. The Llama3 series is additionally tested. Overall, we conduct experiments on Qwen2.5[Team (2024)] (3B, 7B, 32B), DeepSeek-R1-Distill-Qwen (1.5B, 7B, 32B), and Llama[Grattafiori et al. (2024)] (3.1-8B, 3.2-3B).

## 3.1 TRAINING DATASETS AND TRAINING SETTINGS

**Training datasets:** The **LIMO** dataset serves as the starting point dataset. We form 4 dataset variants with Persona and Non-Persona methods (Persona-Similar, Persona-Dissimilar, Persona-Random, and Persona-None). Specifically, for each dataset variant, the original dataset queries have the specific persona strategies applied. These augmented queries, along with the corresponding cot & answers, form the targeted dataset variants. The **OT** dataset is a lower-quality dataset. We sample 1K data points from the math category of the dataset. For simplicity and to address formatting issues (some outputs may not provide answers at all), we only sample from data points that explicitly contain "final answer" and "boxed{" in their outputs. We also form 4 variants with Persona and Non-Persona methods.

**Experiment setups:** We use 360-LlamaFactory[Haosheng Zou & Zhang (2024)] (a variant of LlamaFactory[Zheng et al. (2024)] that supports sequence parallelism) to train our models. We do not use packing and set the maximum token length to 32,768. We fix the number of update steps to approximately 240, the training batch size to approximately 48, and the learning rate to 5e-6 with a cosine learning schedule. We release the training scripts, and more details can be found in the Appendix. We resample examples and regenerate personas, then retrain the corresponding variants to demonstrate the robustness of our method. We use mergekit[Goddard et al. (2024)] to merge model variants. We further report results with a majority vote of three models. Merge and majority vote results can be found in Appendix A.4.

## 3.2 TESTING DATASETS AND EVALUATION SETTINGS

**Testing datasets:** We choose to use AIME24 and AIME25 as our main focuses, since they are widely used to test mathematical performance ceiling, and all other related models use them as the

| Train | Test | LIMO (High Quality) | | | | | | OT (Low Quality) | | | | | |
|---|---|---|---|---|---|---|---|---|---|---|---|---|---|
| | | A24 | A25 | MATH | GPQA | avg | avg2 | A24 | A25 | MATH | GPQA | avg | avg2 |
| N | N | 61.00 | 40.00 | 92.00 | 53.54 | 61.64 | 50.50 | 63.67 | 50.00 | 94.80 | 66.67 | 68.79 | 56.84 |
| | R | 43.33 | 40.00 | 90.20 | 51.01 | 56.14 | 41.67 | 60.00 | 53.33 | 91.20 | 66.67 | 67.80 | 56.67 |
| | S | **63.33** | 40.00 | 89.00 | 50.00 | 60.58 | 51.67 | 60.00 | 56.67 | 90.00 | 64.85 | 67.88 | 58.34 |
| | D | 40.00 | 36.67 | 87.20 | 51.52 | 53.85 | 38.34 | 60.00 | 60.00 | 89.40 | 61.62 | 67.76 | 60.00 |
| R | N | 59.33 | 36.67 | **93.80** | **68.69** | 64.62 | 48.00 | 40.00 | 26.67 | 92.80 | 68.18 | 56.91 | 33.34 |
| | R | **63.33** | 50.00 | 93.20 | 65.66 | 68.05 | 56.67 | 60.00 | 70.00 | 94.80 | 69.70 | 73.63 | 65.00 |
| | S | **63.33** | 53.33 | 93.40 | 66.34 | **69.10** | **58.33** | 73.33 | 53.33 | 94.00 | 67.33 | 72.00 | 63.33 |
| | D | 53.33 | **56.67** | 92.40 | 63.13 | 66.38 | 55.00 | 66.67 | 56.67 | 94.40 | 69.70 | 71.86 | 61.67 |
| S | N | 60.33 | 46.67 | 92.20 | 59.09 | 64.57 | 53.50 | 43.33 | 33.33 | 91.60 | 68.69 | 59.24 | 38.33 |
| | R | 43.33 | 43.33 | 90.20 | 58.08 | 58.74 | 43.33 | 73.33 | 60.00 | 94.40 | 71.72 | 74.86 | 66.67 |
| | S | 60.00 | 50.00 | 92.00 | 56.44 | 64.61 | 55.00 | 66.67 | 66.67 | 94.60 | 69.31 | 74.31 | 66.67 |
| | D | 53.33 | 50.00 | 89.40 | 59.09 | 62.96 | 51.67 | **76.67** | 70.00 | **95.00** | 66.67 | **77.09** | **73.34** |
| D | N | 55.67 | 43.33 | 92.20 | 58.59 | 62.45 | 49.50 | 33.33 | 36.67 | 91.80 | 66.16 | 56.99 | 35.00 |
| | R | 60.00 | 50.00 | 89.00 | 53.03 | 63.01 | 55.00 | 63.33 | 66.67 | 94.60 | **72.22** | 74.21 | 65.00 |
| | S | **63.33** | 46.67 | 90.60 | 57.43 | 64.51 | 55.00 | 70.00 | **73.33** | 94.20 | 67.82 | 76.34 | 71.67 |
| | D | 53.33 | 33.33 | 88.00 | 52.02 | 56.67 | 43.33 | 66.67 | 60.00 | 94.20 | 67.68 | 72.14 | 63.34 |

Table 1: **qwen2.5-32B-Instruct results:** "A24" is short for AIME24, and "A25" is short for AIME25. "Avg" is the average of four dataset results, and "Avg2" only counts AIME24 and AIME25 results. We get the overall best performance with S-D under OT datasets, which is on average 8% higher than the best performance (R-S) under LIMO datasets. The training-testing co-design is more obvious on OT datasets and performs better.

indicator for math reasoning. We also choose MATH500 and GPQA as complementary test sets. For each testing dataset, we form 4 testing variants. Each testing variant has the specific Persona or Non-Persona strategy applied, the same as training dataset variants. For example, for the AIME24 dataset, we create three persona AIME24 datasets and use the original AIME24 dataset as the None-Persona variant.

**Evaluation setups:** Due to resource limits and the complex design of the experiments (for each base model, it will be trained on 2×4 datasets, then each of the trained variants will be tested on 4 test sets. So each base model will have 32 tests on each test set), we used greedy decoding with pass@1 on 32B models. For smaller models, we used pass@1 averaged over 4 samples, the same as in the LIMO paper. We have 8 model variants to test, so we conducted 256 experiment sets.

### 3.3 EXPERIMENT RESULTS EXPLANATIONS

As shown in Table1, the first column (Train) indicates the training strategy used for the model. (N,R,S,D) represents the specific four strategies applied to the SFT datasets. The second column (Test) indicates the testing strategy. The same abbreviations show which specific strategy is applied to the dataset queries. For example, the first line in the table represents a model trained with the Persona-None dataset variant and tested on the Persona-None variant of each test set. "Avg" represents the average of four dataset results, while "Avg2" only counts AIME24 and AIME25. We use two averages due to the empirical findings that applying strategies shows different performance gain patterns on different datasets. We use (training_strategy)-(testing_strategy) to denote the training-testing combinations. For example, "N-" represents the results from a model trained using the Persona-None strategy. "-D" represents the results from testing with Persona-Dissimilar. "S-R" represents training with Persona-Similar and testing on Persona-Random.

## 4 EMPIRICAL FINDINGS ON QUALITY REQUIREMENTS

In this section, we focus on a systematic analysis of our experiment results. **(1)** We first analyze the effect of adding noise to the data. Starting from 32B model results, we then further extend to more models. **(2)** We examine token distributions to understand what happens behind the noise-affected performance patterns. **(3)** We examine the difference between low-quality and high-quality

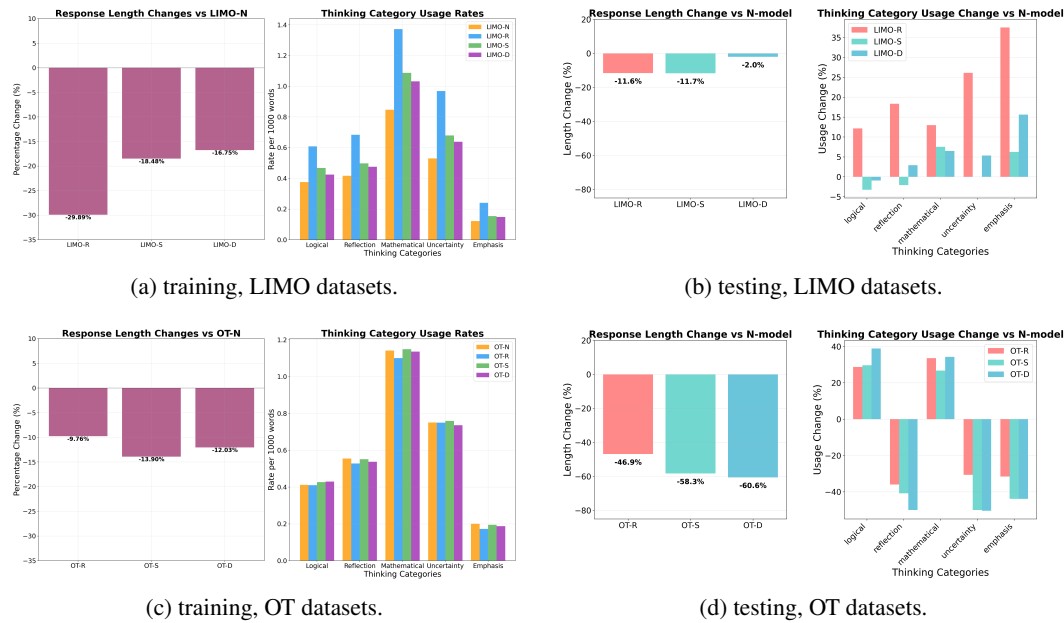

Figure 2: Performance comparison between training and testing strategies of Qwen2.5-32B-Instruct: (a,c) are the training visualizations; (b,d) are the testing visualizations. Applying any strategy during training or testing will shorten the output length, making the reasoning more efficient, which is linked to improved performance.

datasets on highly capable models first. We then further compare results across different model sizes and series.

## 4.1 ADDING NOISE AND IRRELEVANCE INTO QUERIES DURING TRAINING AND TESTING CAN BOOST PERFORMANCE

**Early results on 32B model variants:** We control the noise level by incorporating the relevance of the generated persona contexts to the queries. **The LIMO results** are shown in the left part of Table1. Adding noise (R, D) during training can boost performance for both the performance ceiling compared to adding nothing (N, +8% avg, +8% avg2) and adding relevant information (S, +5% avg, +3% avg2). Applying persona strategies (R,S,D) performs clearly better than applying nothing (N). For training, Persona-S performs the best, while Persona-R can perform comparably. Applying personas (R,S,D) during testing can also provide a higher performance ceiling than applying nothing. **The OT results** are shown in the right part of Table1. Similar results hold, while the difference between applying or not applying strategies becomes greater. Applying strategies consistantly during training and testing ([R,S,D]-[R,S,D]) shows much better results than inconsistently applying ([R,S,D]-N: +18% avg, +35% avg2; N-[R,S,D]: +10% avg, +13% avg2; N-N: +8% avg, +16% avg2)

Notably, models trained with any persona strategy (R, S, D) and tested on any persona strategy consistently achieve high scores on both AIME24 and AIME25, This phenomenon underscores the importance of employing persona strategies throughout both training and testing phases.

Integrating irrelevant and noisy information during training and testing does not inherently compromise model performance. Testing with the Persona-Similar strategy yields the most consistent results, while incorporating noisy information can show comparable or higher results. Training with noise (R, D) can boost performance in many cases compared to applying nothing or adding relevant information. Furthermore, our findings reveal that training with any persona strategy combined with testing using any configuration ([R,S,D]-[R,S,D]) consistently produces performance improvements. Conversely, the absence of a persona strategy during either training or testing significantly diminishes performance. Taken together, the consistent application of persona strategies across both

training and testing phases emerges as the critical factor. We refer to this phenomenon as **training-testing co-design**.

**Extended validations across 8 models:** We conducted a comprehensive comparison across 8 distinct models, with detailed results presented in Appendix A.7. Training with noisy information generally yields superior performance: in 9 of 16 experiments, the R or D strategies produced the best results; in 4 of 16 experiments, strategy S achieved optimal performance; and in the remaining 3 experiments, the N training strategy performed best. Regarding testing approaches, strategy S demonstrates the most consistent performance, although strategies D or R frequently exhibit superior results in most scenarios. It is notably rare for strategy N to achieve optimal testing performance. Overall, training and testing with noisy information consistently enhances performance. Furthermore, a substantial performance gap exists depending on whether persona strategies are comprehensively applied to queries. Configurations (N-, -N) consistently demonstrate significantly inferior performance compared to ([R,S,D]-[R,S,D]). And this disparity is becoming more pronounced as model size increases within the same series.

**Key Insight 1:** The integration of irrelevant and noisy information into queries during training and testing can boost model performance. Consistent application of persona contexts throughout both training and testing phases ([R,S,D]-[R,S,D]) is crucial for eliciting substantial performance improvements, particularly on OT datasets when models possess sufficient capacity. We designate this phenomenon as **training-testing co-design**.

## 4.2 APPLYING PERSONA CONTEXTS MAKE REASONING MORE EFFICIENT UNDER TOKEN DISTRIBUTION ANALYSIS

We employ token distribution analysis to elucidate the underlying factors driving performance outcomes. Specifically, we categorize thinking tokens into five distinct groups: logic, reflection, mathematics, uncertainty, and emphasis. Detailed information regarding these categories is provided in Appendix A.3.

**Training Strategy Analysis:** As illustrated in 2 (a), a distinct response length variation emerges based on the implementation of persona strategies (N vs. [R,S,D]) within LIMO datasets. Concurrently, the proportions of thinking tokens across different categories in the generated chains-of-thought remain comparable or increase. The application of training strategies enables more compact reasoning, articulates thinking tokens more effectively, and enhances overall performance. Analogous results are observed in the OT variants presented in (c). These findings hold consistently across qwen2.5-7B-Instruct and DeepSeek-Distill-Qwen-7B (OT variants) in 4, as well as DeepSeek-Distill-Qwen32B (OT) variants in 3. Notably, the LIMO variants of DeepSeek-Distill-Qwen (both 7B and 32B) exhibit increased response lengths. Analysis of their performance metrics reveals that optimal results are achieved through training with the Persona-N strategy. In summary, superior performance correlates with a more efficient reasoning process. The concatenation of personas to queries during training demonstrably alters model reasoning behaviors.

**Testing Strategy Analysis:** A consistent pattern of response length reduction emerges in testing strategy comparisons. As demonstrated in 2 (b)(d) and 3 (d), superior model performance correlates with more efficient reasoning processes. Notably, LIMO variants of DeepSeek-Distill-Qwen (32B) exhibit increased response lengths, as their optimal performance occurs under Persona-None testing conditions, which remains consistent with our empirical findings.

**Key Insight 2:** Applying persona strategies throughout both training and testing phases demonstrates an overall enhancement in reasoning process efficiency. Shorter response lengths link to higher model performance.

## 4.3 LOWER-QUALITY DATASETS CAN PERFORM BETTER ON MATH REASONING TASKS

**32B model results:** As demonstrated in Table 1, we first compare the results between LIMO and OT datasets under the Qwen2.5-32B-Instruct model. OT datasets exhibit superior performance, achieving approximately 8% higher avg than LIMO datasets. When examining AIME24 and AIME25 specifically (avg2), the performance advantage increases to 15%. Their result patterns are different. For LIMO variants, models trained on Persona-None or Persona-Similar experience performance degradation when exposed to noise or irrelevant information. Conversely, models trained

on Persona-Dissimilar and Persona-Random can achieve further performance improvements. The maximum performance is achieved with Persona-Random variants: 63.3% pass@1 on AIME24 and 53.3% pass@1 on AIME25. Furthermore, implementing persona strategies during both training and testing substantially enhances performance on AIME24 while maintaining comparable results on AIME25. The performance patterns differ significantly for OT datasets, where we achieve 76.7% pass@1 on AIME24 and 70% pass@1 on AIME25, representing an average improvement of 15%. When trained on Persona-None, all test persona strategies demonstrate superior results on AIME25 compared to Persona-None testing. For DeepSeek-R1-Distil-Qwen-32B models, results are presented in Table 8. LIMO datasets achieve their highest scores on N-N combinations. While OT can have slightly higher performance on both avg and avg2. Beyond this atypical behavior, OT datasets demonstrate a clear advantage over LIMO results. On average, training on OT datasets yields approximately 10% higher performance.

**Model Size Effect Analysis:** For very small models with limited capabilities, high-quality datasets demonstrate superior performance. However, as model size increases, this performance gap diminishes, and at the 32B scale, lower-quality datasets can actually elevate the performance ceiling substantially higher. Through careful examination of dataset behaviors, we observe that lower-quality datasets facilitate improved learning of challenging questions, regardless of model size. Nevertheless, the average score is dominated by easier datasets (MATH) when models are small and less capable. High-quality datasets exhibit a distinct advantage in introducing knowledge for such easy questions. As model capabilities expand, the performance gap on MATH datasets narrows progressively. Consequently, the average score becomes dominated by challenging datasets, where lower-quality datasets demonstrate a clear advantage. Lower-quality datasets can extend the performance ceiling by excelling on challenging tasks, while maintaining comparable performance on easy questions.

**Key Insight 3**: Low-quality datasets can help improve performance on hard questions much more effectively, as the model becomes larger and more capable, while high-quality datasets have an advantage in improving performance on easy questions for small models. Applying training-testing co-design is especially important for low-quality datasets to get the performance boost.

## 5 INPUT-TIME SCALING AND ITS EFFECTIVENESS

Input-Time Scaling is an empirically efficient approach, where we train sufficiently capable models on very small and low-quality datasets. It requires us to apply persona strategies during both training and testing, and in return, it can achieve 27% higher absolute scores on hard AIME problems, compared to applying no strategies at all. This effectiveness can make very small and low-quality datasets much more beneficial and free us from the expensive and time-consuming data curation process.

A comparison of results between our method and other models can be found in Table2. We first compare the performance of Qwen2.5-32B-based models: **(1)** Compared to LIMO and S1, we use unfiltered datasets with similar dataset sizes (1K). Instead of focusing on quality, difficulty, and diversity, we focus on automatically adding noise and irrelevance using pure meta-cognition. Our final results, based on the same Qwen2.5-32B-Instruct models, achieve more than 10% improvements on AIME24. More importantly, we achieve 20% improvement on AIME25, successfully reducing the performance gap between the similar difficulty problems. **(2)** Compared to OpenThinker2-32B, we used the same base model, Qwen2.5-32B-Instruct, and the same SFT process, but with 1000x less data by randomly sampling from their math subset examples. The performance on AIME24 is the same, but we successfully achieve 22% higher performance on AIME25. **(3)** Compared to MiroMind-M1-RL-32B (62K RL data) and Skywork-OR1-32B-preview (124K RL data), both equipped with multi-stage training, our method works directly with only 1K SFT data. Starting from the same DeepSeek-R1-Distill-Qwen-32B, our method achieves an average 13% performance gain on AIME24 and AIME25. **(4)** Compared to QwQ-32B (pretraining & SFT & RL) and DeepSeek-R1-Distill-Qwen-32B (800K SFT data), our method is comparatively much easier to use with a remarkably small SFT dataset. We achieve 90% on AIME24 and 80% on AIME25, which is 10% higher than QwQ-32B and 20% higher than the other. In summary, our method achieves state-of-the-art performance on AIME24 and AIME25 while using only 1K SFT data and focusing on adding noisy and irrelevant information during both training and testing. **(5)** When we examine

| Model | data | train | ITS | A24 | A25 | MATH | GPQA |
|---|---|---|---|---|---|---|---|
| LIMO-32B | 1k | SFT | - | 56.7 | 49.3 | 86.6 | 66.7 |
| S1-32B | 1k | SFT | - | 36.0 | 25.3 | 84.8 | 59.6 |
| S1.1-32B | 1k | SFT | - | 64.7 | 49.3 | 89.0 | 63.6 |
| OpenThinker-32B | 114k | SFT | - | 68.0 | 49.3 | 90.6 | 63.5 |
| OpenThinker2-32B | 1000K | SFT | - | 76.7 | 58.7 | 90.8 | 64.1 |
| Skywork-OR1-32B-Preview | 124K | RL | - | 77.1 | 68.2 | 97.5 | - |
| MiroMind-M1-RL-32B | 62K | RL | - | 77.5 | 65.6 | 96.4 | - |
| QwQ-32B | - | RL | - | 79.5 | 69.5 | 98.0 | - |
| Qwen3-32B | - | - | - | 81.4 | 72.9 | 97.2 | - |
| Qwen3-235B-A22B | - | - | - | 85.7 | 81.5 | 98.0 | - |
| OpenAI-o1 | - | - | - | 74.3 | 79.2 | 96.4 | - |
| OpenAI-o3-mini(medium) | - | - | - | 79.6 | 74.8 | 98.0 | - |
| Grok-3-Beta(Think) | - | - | - | 83.9 | 77.3 | - | - |
| DeepSeek-R1 | - | - | - | 79.8 | 70.0 | - | - |
| DeepSeek-Llama-70B | 800K | SFT | - | 70.0 | 56.3 | 94.5 | - |
| DeepSeek-Qwen-32B | 800K | SFT | - | 70.8 | 52.1 | - | - |
| DeepSeek-Qwen-32B (Greedy Decoding) | 800K | SFT | - | 56.7 | 40.0 | 90.0 | 50.0 |
| Qwen2.5-32B-Instruct | 1K | SFT | S-D | 76.7 | 70 | 95.0 | 69.7 |
| | | | D-S | 70 | 73.3 | 94.2 | 70.2 |
| | | | Merge | 70 | 76.7 | 95.0 | 68.7 |
| | | | Majority Vote (S-D D-S R-R) | 73.3 | 80.0 | 94.2 | 70.2 |
| | | | Majority Vote (S-D D-S R-S) | 76.7 | 76.7 | 95.2 | 70.8 |
| DeepSeek-Qwen-32B | 1K | SFT | D-S | 86.7 | **80.0** | 95.2 | 70.2 |
| | | | S-S | **90.0** | 73.3 | 94.4 | 67.8 |
| | | | R-D | 83.3 | 73.3 | 96.2 | 66.2 |
| | | | Majority Vote (D-S S-S S-S) | 90.0 | 80.0 | 94.8 | 67.8 |
| | | | Majority Vote (R-S D-R S-S) | 90.0 | 80.0 | 96.4 | 68.2 |

Table 2: **Comparing the Performance of SOTA models** This table comparison showing pass@1 percentages across different models. We annotate the training dataset sizes, and their training methods to provide more details on how our method excels. The rows with ITS column filtered are our models under specific strategies.

larger models, our method can even surpass DeepSeek-R1[Guo et al. (2025)], OpenAI-o1[OpenAI (2024)], OpenAI-o3-mini (medium)[OpenAI (2025)], Grok-3-Beta[Abramov et al. (2025)], and Qwen3-235B-A22B[Yang et al. (2025)] on AIME24, and is comparable to them on AIME25. With such a small and old base model, our method achieves higher performance compared to these new-structure, large-scale training models.

**Key Insight 4**: Input-Time Scaling can outperform similar-size models that include significantly more training data of higher quality, and it can also outperform much larger reasoning models.

# 6 CONCLUSION

In this work, we focus on systematically relaxing the quality requirements of the "Less is More" phenomenon along two dimensions. We first add noise to queries during both training and testing by controlling the relevance to the original queries for different noise levels. We find that adding noise does not necessarily degrade performance and can even improve it. During our investigation of token distributions, applying strategies makes the reasoning process more efficient and thus improves performance. For datasets of different quality, high-quality datasets excel when the model is weak and focuses on easy questions. As models become increasingly capable, low-quality datasets can

actually boost overall performance, especially on hard questions. Overall, the quality requirement does not hold across our experiments. From the previous analysis, we discover the training-testing co-design phenomenon: adding any context during training and testing boosts performance. Thus, we frame our method as applying small, low-quality datasets to sufficiently capable models with persona strategies during both training and testing. We name the overall design Input-Time Scaling, under which we achieve state-of-the-art performance on 32B Qwen2.5 model variants. With only 1K low-quality training examples, we reach 76.7% pass@1 on both AIME24 and AIME25. When we start from DeepSeek-R1-Distill-Qwen-32B, we achieve 90% and 80% pass@1 on AIME24 and AIME25, respectively. To facilitate reproducibility and further research, we will open-source our datasets, data pipelines, evaluation results, and checkpoints. Our work serves as a new starting point for investigating training-testing co-design and data efficiency and efficacy.

## REPRODUCIBILITY STATEMENT

You can find the train&test sets curation details in 2 part. The specific prompts can be found in the APPENDIX A.6. Our complete pipeline is to curate the corresponding train&test sets, using SFT as shown in the 3 part, and evaluate using the curated test sets. There is no filtering during the whole process, and all samples are selected randomly as mentioned.

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

# A APPENDIX

## A.1 RELATED WORKS

Current large language models (LLMs) (Guha et al. (2025); Li et al. (2025c); Guo et al. (2025); He et al. (2025)) have achieved impressive performance in math and other reasoning tasks. They are usually post-trained on carefully curated large-scale datasets (data & training scaling), and undergo a two-stage training pipeline. It requires different intuitive heuristics (inductive biases) to guide filtering(Havrilla et al. (2024); Zhang et al. (2024); Ye et al. (2025); Li et al. (2025a)). Quality and diversity are important for LLMs, but defining quality is non-trivial; diversity is relatively more objective (Havrilla et al. (2024); Zhang et al. (2024); Guha et al. (2025)). Some methods (Ye et al. (2025); Guha et al. (2025); Snell et al. (2024); Muennighoff et al. (2025); Balachandran et al. (2025)) consider difficulty a more proven method to select data. If the query contains more steps, needs more information and has a lower pass rate, the query is more difficult. The essence of difficulty relies on the possible action space instead of the quality of the data. Recently, LIMO(Ye et al. (2025)), s1(Muennighoff et al. (2025)) and some works adapt the Less is More hypothesis to reasoning. They show that using a small set of high-quality and precise reasoning demonstrations is enough to obtain strong results. However, some works(Sun et al. (2025)) point to their comparably restrained ceiling to scaling the dataset sizes.

(Jang et al. (2025); Schmied et al. (2025); Gekhman et al. (2025))There is a gap between model performance and model knowledge. Currently, there emerges one direction to close this gap. Instead of training or data scaling, they do reasoning in test time (inference time scaling) using more computational resources to generate the results. (Snell et al. (2024)) find that scaling test-time computation can be more effective than scaling the size or training of the models with a similar computational budget. There are depth-wise scaling methods, such as CoT(Wei et al. (2022)), reflection (Guo et al. (2025); Bensal et al. (2025)), to carefully generate reasoning trajectories. It can recover from early

errors, making reasoning processes more stable. On the other hand, width-wise methods, like self-consistency(Wang et al. (2022)) and BoN, can invoke inner diversity to creatively explore the ability ceiling. Further mixing multiple outputs can further improve performance(Song et al. (2025); Li et al. (2025b)). However, knowing when and how to scale remains an open question(Wang et al. (2025a); Zeng et al. (2025)).

## A.2 THE USE OF LARGE LANGUAGE MODELS (LLMS)

We used Overleaf(Overleaf (2025)) to improve writing and Cursor Cursor (2025) to help code.

## A.3 TOKEN CATEGOREIS

We categorize tokens into five groups based on their semantic roles:

**Logical:** logical, reasoning, therefore, consequently, implies, follows that, deduce, infer, conclusion, premise, argument, valid, sound, if...then, necessary, sufficient, contradiction, fallacy

**Reflection:** reflect, think about, reconsider, rethink, looking back, in retrospect, upon reflection, wait, hold on, on second thought, reconsidering

**Mathematical:** calculate, compute, equation, formula, theorem, proof, mathematical, numerical, algebraic, geometric, arithmetic, derivative, integral, function, variable, constant, solve, solution, answer is, equals, sum, product

**Uncertainty:** maybe, perhaps, possibly, might, could be, uncertain, not sure, unclear, ambiguous, doubt, question, probably, likely, seems

**Emphasis:** definitely, certainly, absolutely, clearly, obviously, undoubtedly, without doubt, surely, indeed, in fact, especially, particularly, notably, importantly, key

## A.4 MORE EXPERIMENTS

**Simply scaling the dataset size**

| Train | Test | OT-15k Dataset | | | | | | OT-1k Dataset | | | | | |
|---|---|---|---|---|---|---|---|---|---|---|---|---|---|
| | | A24 | A25 | MATH | GPQA | Avg | Avg2 | A24 | A25 | MATH | GPQA | Avg | Avg2 |
| N | N | 46.67 | 50.00 | 89.80 | 52.02 | 59.62 | 48.34 | 63.67 | 50.00 | 94.80 | 66.67 | 68.79 | 56.84 |
| | R | 40.00 | 36.67 | 79.20 | 53.54 | 52.35 | 38.34 | 60.00 | 53.33 | 91.20 | 66.67 | 67.80 | 56.67 |
| | S | 50.00 | 30.00 | 79.80 | 43.56 | 50.84 | 40.00 | 60.00 | 56.67 | 90.00 | 64.85 | 67.88 | 58.34 |
| | D | 53.33 | 50.00 | 79.20 | 45.45 | 56.99 | 51.67 | 60.00 | 60.00 | 89.40 | 61.62 | 67.76 | 60.00 |
| R | N | 45.67 | 23.33 | 88.40 | 55.56 | 53.24 | 34.50 | 40.00 | 26.67 | 92.80 | 68.18 | 56.91 | 33.34 |
| | R | 50.00 | 43.33 | 92.00 | 44.44 | 57.44 | 46.67 | 60.00 | **70.00** | 94.80 | 69.70 | 73.63 | 65.00 |
| | S | 63.33 | 53.33 | 92.40 | 45.05 | 63.53 | 58.33 | **73.33** | 53.33 | 94.00 | 67.33 | 72.00 | 63.33 |
| | D | 60.00 | 66.67 | 91.20 | 47.98 | 66.46 | 63.34 | 66.67 | 56.67 | 94.40 | 69.70 | 71.86 | 61.67 |
| S | N | 52.33 | 30.00 | 88.80 | 56.57 | 56.93 | 41.17 | 43.33 | 33.33 | 91.60 | 68.69 | 59.24 | 38.33 |
| | R | 63.33 | 56.67 | 89.00 | 44.44 | 63.36 | 60.00 | 73.33 | 60.00 | 94.40 | **71.72** | 74.86 | 66.67 |
| | S | 53.33 | 56.67 | 87.80 | 46.53 | 61.08 | 55.00 | 66.67 | 66.67 | 94.60 | 69.31 | 74.31 | 66.67 |
| | D | 46.67 | 50.00 | 89.60 | 35.35 | 55.41 | 48.34 | **76.67** | **70.00** | 95.00 | 66.67 | 77.09 | 73.34 |
| D | N | 46.33 | 33.33 | 88.40 | 51.52 | 54.90 | 39.83 | 33.33 | 36.67 | 91.80 | 66.16 | 56.99 | 35.00 |
| | R | 50.00 | 43.33 | 91.40 | 45.45 | 57.55 | 46.67 | 63.33 | 66.67 | 94.60 | **72.22** | 74.21 | 65.00 |
| | S | 66.67 | 46.67 | 90.20 | 45.54 | 62.27 | 56.67 | **70.00** | **73.33** | 94.20 | 67.82 | 76.34 | 71.67 |
| | D | 66.67 | 60.00 | 90.80 | 43.43 | 65.23 | 63.34 | 66.67 | 60.00 | 94.20 | 67.68 | 72.14 | 63.34 |

Table 3: Combined OT Dataset Results. We found that scaling the dataset size to 15k behaves much worse than only using 1K data. With on average 15% performance down.

There are some works (Sun et al. (2025)) arguing simply scaling the dataset size can further improve the model performance. Starting from this view, we scale our used OT datasets, and we find that scaling the dataset size to 15k behaves much worse than only using 1K data. The results can be found in table 3. For OT-15k results, there is no clear best strategy combination. The best performance

is obtained using **D-D** (66.7% 60.0%) on AIME24 and AIME25, which is on average about 10% worse than OT results. And if we consider the average of four datasets, it is on average 15% worse.

**Merge & Majority vote**

The merge results can be found in table 4. Merge models of Qwen2.5-32B-Instruct (S, D, R) achieve higher and more stable performance. They can achieve 76.7% on AIME24 and 76.7% on AIME25. The results of the majority vote can be found in table 5 and 4. Using different strategies, the model achieves 80.0% on AIME24 and 76.7% on AIME25 with Qwen2.5-32B-Instruct. Starting from DeepSeek-R1-Distill-Qwen-32B, the model achieves 90.0% on AIME24 and 80.0% on AIME25.

Table 4: Merge results

| Train | Test | AIME24 | AIME25 | MATH | GPQA |
|---|---|---|---|---|---|
| | N | 40.00 | 43.33 | 91.40 | 65.66 |
| Merge | R | 70.00 | **76.67** | 95.00 | 70.71 |
| | S | 70.00 | 70.00 | 94.20 | **73.76** |
| | D | **76.67** | 63.33 | 93.80 | 68.18 |

Table 5: Majority vote results

| Majority vote | AIME24 | AIME25 | MATH |
|---|---|---|---|
| S-D D-S R-R | 0.733 | **0.800** | 0.942 |
| S-D D-S R-S | **0.767** | 0.767 | 0.952 |
| S-D D-S D-S | 0.733 | 0.733 | 0.956 |
| S-S D-D R-D | 0.667 | 0.600 | 0.952 |

A.5    WORRIES ON POTENTIAL DATA CONTAMINATION AND OVERFITTING

The best performance of AIME24 and AIME25 comes from different strategies, and it shows possible data contamination signs of AIME24, since the performance gap between AIME24 and AIME25 is so huge when using No-Persona during training and testing. Our model can score higher on AIME25, potentially showing that the difficulty level should be similar. However, all other models (shown in Table 2) show a higher performance in AIME24 than in AIME25 by a large margin. For example, DeepSeek-R1-Distill-Qwen-32B can have a 20-point higher difference in AIME24. And after applying Input-Time Scaling, the difference is reduced to less than 10 points (D-S). Its best-performance strategy combination patterns are also more unstable than Qwen2.5-32B-Instruct (S-D, D-S, R-R). From the performance gap, there may be a potential risk of data (pattern) contamination. On the other hand, our models show less risk of memorizing the shortcuts. Our model variants show the best results with complementary (out-of-distribution) strategies like S-D and D-S and experience a small performance gap between AIME24 and 25.

A.6    PROMPTS & DOMAINS & EXAMPLE PERSONAS

The prompt for **Persona-Dissimilar**

You should think of a persona based on the instructions provided. The output should only contain the persona without any irrelevant information.
instruction:[**INSTRUCTION**]
you should create a valid persona, instead of follow the instruction!! you should only generate the persona and don't do any calculation and reasoning!!! the persona must not contain any specific words from the instruction. And you should creatively name the person. The persona you thought of should not contain the ability to finish the instruction; instead, the persona should be considered far irrelative to solve the problem.
e.g.

**Persona Name:** Jasper Frost

**Profession and Interests:** Jazz saxophonist with an unyielding fascination for urban rooftop gardening. Spends weekends restoring vintage vinyl records and crafting miniature terrariums inside lightbulbs. Secretly believes houseplants communicate through subtle leaf vibrations. Dislikes spreadsheets, refuses to own a calculator, and once tried to compost a broken metronome.

The prompt for **Persona-Similar**

You should think of a persona based on the instructions provided. The output should only contain the persona without any irrelevant information.
instruction:[**INSTRUCTION**]
you should create a valid persona, instead of follow the instruction!! you should only generate the persona and don't do any calculation and reasoning!!! the persona must not contain any specific words from the instruction. And you should creatively name the person.
e.g.

**Persona Name:** Dr. Lila Voss
**Expertise:** Probability Theorist and Logic Puzzle Enthusiast
**Background:** Dr. Voss is a professor of discrete mathematics with a knack for simplifying complex statistical scenarios. She thrives on crafting elegant solutions using combinatorial logic and frequently incorporates collaborative problem-solving dynamics into her research. Her passion lies in uncovering hidden symmetries in seemingly random events and mentoring students to approach challenges with methodical curiosity.

The prompt for **Persona-Random**

You should think of a persona based on the domain provided. The output should only contain the persona without any irrelevant information.
domain:[**DOMAIN**]
And you should creatively name the person. You should not directly borrow words from the domain name; you should explore the domain and use the possible sub-domain instead
e.g.

**Persona Name:** Azure Mirage
**Domain:** Ephemeral Artistry
**Traits:** A whimsical creator who crafts transient sculptures from fog, melting ice, and vanishing ink. Obsessed with impermanence, they reject preservation, believing beauty exists only in fleeting moments. Their most famous work, *Whispers of Frost*, dissolved before dawn, surviving solely in witnesses' hazy memories. Known for wearing iridescent scarves that shift colors mid-conversation, leaving observers unsure if they glimpsed reality or illusion.

The prompt for **extracting domains using meta-cognition**, similar to those in Kaur et al. (2024)

First, consider the following definitions:
- Knowledge: A set of information, facts, descriptions, and theories acquired through education of experience.
- Topic: A subset of knowledge that focuses on a particular area or subject.
Now, I am creating a dataset of engaging human knowledge that demonstrate the capabilities and value of human beings across the knowledge we perceive. Please help me generate high-quality and well coverages topics (in snake case) for this dataset. You should provide the answers with the format answer-explanation.
At last, you need give me an analyze on the topics you provided, and explain why these topics together can be a whole representation of human knowledge.

The **domains** extracted

1. **Natural Sciences**: Physics, chemistry, biology, earth sciences, and astronomy — foundational understanding of the physical universe and its laws.

2. **Formal Sciences**: Mathematics, logic, statistics, and theoretical frameworks (e.g., computational theory) — tools for abstract reasoning and problem-solving.

3. **Applied Sciences & Technology**: Engineering, robotics, AI, nanotechnology, and sustainable tech — practical innovations that reshape society and industry.

4. **Social Systems**: Governance, economics, law, education, and urbanization — structures that organize human collaboration and societal progress.

5. **Philosophy & Ethics**: Metaphysics, epistemology, moral philosophy, and political theory — frameworks for critical thinking and value-driven decision-making.

6. **History & Cultural Evolution**: Civilizations, conflicts, technological diffusion, and cultural exchange — lessons from humanity's collective past.

7. **Environmental Systems**: Ecology, climate science, sustainability, and resource management — interdependence of humans and planetary health.

8. **Health & Medicine**: Disease prevention, genetics, mental health, and healthcare systems — advancing physical and psychological well-being.

9. **Language & Communication**: Linguistics, semiotics, translation, and digital media — tools for expressing ideas and bridging cultural divides.

10. **Creative Expression**: Visual arts, literature, music, culinary arts, and design — cultural identity and innovation through aesthetic forms.

11. **Space & Exploration**: Astrophysics, planetary science, and interstellar travel — expanding humanity's physical and intellectual frontiers.

12. **Psychology & Behavior**: Cognition, emotion, social dynamics, and neurodiversity — understanding individual and collective human behavior.

13. **Digital Revolution**: Cybersecurity, blockchain, quantum computing, and data science — adaptation to the transformative power of information.

14. **Ritual& Belief Systems**: Religion, spirituality, festivals, and rites of passage — symbolic practices that shape meaning and community.

## A.7 TABLES & CHARTS

This section contains training-testing combination results of qwen2.5 (3B, 7B, 32B), deepseek-r1-distill (1.5B, 7B, 32B), and Llama3 (3B, 8B). The charts contains the results of 7b and 32B models.

| Train | Test | LIMO (High Quality) | | | | | | OT (Low Quality) | | | | | |
|---|---|---|---|---|---|---|---|---|---|---|---|---|---|
| | | A24 | A25 | MATH | GPQA | avg | avg2 | A24 | A25 | MATH | GPQA | avg | avg2 |
| N | N | 7.50 | 6.67 | 69.40 | 5.05 | 22.15 | 7.08 | 10.00 | 11.67 | 42.80 | 1.01 | 16.37 | 10.84 |
| | R | 11.67 | 11.67 | 67.00 | 3.03 | 23.34 | 11.67 | 14.17 | 14.17 | **53.60** | **2.53** | **21.12** | 14.17 |
| | S | 10.83 | 5.83 | 72.00 | 1.49 | 22.54 | 8.33 | 17.50 | 11.67 | 48.00 | 1.49 | 19.66 | 14.59 |
| | D | 9.17 | 12.50 | 72.00 | 3.54 | 24.30 | 10.84 | 11.67 | **15.83** | 50.00 | 2.02 | 19.88 | 13.75 |
| R | N | 13.33 | 12.50 | 70.80 | **7.07** | 25.93 | 12.91 | 12.50 | 13.33 | 41.60 | 0.51 | 16.98 | 12.91 |
| | R | 10.83 | 8.33 | 70.80 | 5.05 | 23.75 | 9.58 | 10.83 | 10.00 | 45.00 | **2.53** | 17.09 | 10.41 |
| | S | 14.17 | 9.17 | **74.80** | 4.95 | 25.77 | 11.67 | 14.17 | 12.50 | 40.60 | 0.00 | 16.82 | 13.34 |
| | D | **16.67** | 10.00 | 74.20 | 5.05 | 26.48 | 13.33 | 15.00 | 12.50 | 42.00 | 1.01 | 17.63 | 13.75 |
| S | N | 8.33 | 10.00 | 71.00 | 3.54 | 23.22 | 9.16 | 15.00 | 14.17 | 42.40 | 1.01 | 18.14 | 14.59 |
| | R | 12.50 | 10.00 | 73.40 | 6.57 | 25.62 | 11.25 | 14.17 | 15.00 | 43.60 | 0.51 | 18.32 | 14.59 |
| | S | 8.33 | 10.83 | **74.80** | 2.97 | 24.23 | 9.58 | **21.67** | 10.83 | 38.20 | 0.50 | 17.80 | **16.25** |
| | D | 15.00 | **18.33** | 73.40 | 5.05 | **27.95** | **16.66** | 15.00 | 13.33 | 47.40 | 0.51 | 19.06 | 14.16 |
| D | N | **16.67** | 15.00 | 71.20 | 5.56 | 27.11 | 15.83 | 15.00 | 10.83 | 43.00 | 0.51 | 17.34 | 12.91 |
| | R | 14.17 | 7.50 | 73.00 | 6.06 | 25.18 | 10.84 | 17.50 | 9.17 | 42.20 | 1.01 | 17.47 | 13.34 |
| | S | 10.00 | 15.00 | 73.00 | 5.45 | 25.86 | 12.50 | 8.33 | 9.17 | 42.60 | 0.99 | 15.27 | 8.75 |
| | D | 14.17 | 10.83 | 73.80 | 6.06 | 26.21 | 12.50 | 15.83 | 12.50 | 44.40 | 1.52 | 18.56 | 14.16 |

Table 6: **DeepSeek-R1-Distill-Qwen-1.5B:** "A24" is short for AIME24, and "A25" is short for AIME25. "Avg" is the average of four dataset results, and "Avg2" only counts for AIME24 and AIME25 results. We get the overall best performance with S-D under LIMO datasets. Avg2 is almost the same, and OT can even have a better performance, however, Avg is dominated by the MATH score, and LIMO perform much better than OT.

| Train | Test | LIMO (High Quality) | | | | | | OT (Low Quality) | | | | | |
|---|---|---|---|---|---|---|---|---|---|---|---|---|---|
| | | A24 | A25 | MATH | GPQA | avg | avg2 | A24 | A25 | MATH | GPQA | avg | avg2 |
| N | N | 64.17 | 42.50 | 92.40 | 32.32 | **57.85** | 53.34 | 56.67 | 38.33 | 84.40 | 18.18 | 49.39 | **47.50** |
| | R | 60.83 | 44.17 | 88.00 | 23.23 | 54.06 | 52.50 | 52.50 | 38.33 | 88.40 | 20.20 | 49.86 | 45.41 |
| | S | 55.83 | 42.50 | 90.00 | 23.76 | 53.02 | 49.16 | 50.00 | 39.17 | 84.40 | 19.80 | 48.34 | 44.59 |
| | D | 60.83 | 45.83 | 88.40 | 24.24 | 54.83 | 53.33 | 51.67 | **40.83** | 85.60 | 21.72 | 49.95 | 46.25 |
| R | N | 60.83 | 38.33 | 91.80 | **32.83** | 55.95 | 49.58 | 49.17 | 34.17 | 85.40 | **27.27** | 49.00 | 41.67 |
| | R | 57.50 | 45.00 | **93.00** | 27.27 | 55.69 | 51.25 | 46.67 | 37.50 | 88.00 | 19.70 | 47.97 | 42.09 |
| | S | 57.50 | **48.33** | 90.80 | 29.70 | 56.58 | 52.91 | 47.50 | 34.17 | 84.40 | 23.76 | 47.46 | 40.84 |
| | D | **65.00** | 42.50 | 91.20 | 30.30 | 57.25 | **53.75** | 45.83 | 34.17 | 86.20 | 25.25 | 47.86 | 40.00 |
| S | N | 55.83 | 45.00 | 90.60 | 29.80 | 55.31 | 50.41 | 49.17 | 30.83 | 89.40 | 23.23 | 48.16 | 40.00 |
| | R | 55.83 | 40.83 | 91.20 | 27.27 | 53.78 | 48.33 | 52.50 | 32.50 | 88.40 | 23.74 | 49.29 | 42.50 |
| | S | 55.00 | 45.00 | 92.00 | 23.27 | 53.82 | 50.00 | 47.50 | 35.83 | **89.60** | 26.73 | 49.91 | 41.66 |
| | D | 59.17 | 44.17 | 90.60 | 27.78 | 55.43 | 51.67 | 47.50 | 36.67 | 87.60 | 23.23 | 48.75 | 42.09 |
| D | N | 58.33 | 44.17 | 91.00 | 28.28 | 55.45 | 51.25 | 54.17 | 35.83 | 87.20 | 24.24 | **50.36** | 45.00 |
| | R | 61.67 | 41.67 | **93.00** | 24.75 | 55.27 | 51.67 | 48.33 | 37.50 | 86.60 | 22.22 | 48.66 | 42.91 |
| | S | 55.83 | 45.83 | 91.20 | 24.26 | 54.28 | 50.83 | 47.50 | 33.33 | 86.40 | 18.81 | 46.51 | 40.41 |
| | D | 55.00 | 44.17 | 89.80 | 26.26 | 53.81 | 49.59 | 46.67 | 35.83 | 88.20 | 23.74 | 48.61 | 41.25 |

Table 7: **DeepSeek-R1-Distill-Qwen-7B:** "A24" is short for AIME24, and "A25" is short for AIME25. "Avg" is the average of four dataset results, and "Avg2" only counts for AIME24 and AIME25 results. We get the overall best performance with S-D under LIMO datasets. The difference on percentage is smaller than that on 1.5B models.

| Train | Test | LIMO (High Quality) | | | | | | OT (Low Quality) | | | | | |
|---|---|---|---|---|---|---|---|---|---|---|---|---|---|
| | | A24 | A25 | MATH | GPQA | avg | avg2 | A24 | A25 | MATH | GPQA | avg | avg2 |
| N | N | **86.67** | **73.33** | 94.80 | 66.16 | **80.24** | **80.00** | 56.67 | 40.00 | 90.00 | 50.00 | 59.17 | 48.34 |
| | R | 66.67 | 60.00 | 89.20 | 65.15 | 70.25 | 63.34 | 63.33 | 46.67 | 81.40 | 56.57 | 61.99 | 55.00 |
| | S | 63.33 | 66.67 | 91.20 | 67.33 | 72.13 | 65.00 | 63.33 | 40.00 | 83.20 | 57.92 | 61.11 | 51.66 |
| | D | 76.67 | 60.00 | 85.40 | 66.67 | 72.19 | 68.34 | 43.33 | 36.67 | 76.00 | 56.06 | 53.02 | 40.00 |
| R | N | 76.67 | 70.00 | 94.40 | **71.72** | 78.20 | 73.34 | 70.00 | **80.00** | 95.60 | 70.20 | 78.95 | 75.00 |
| | R | 66.67 | 63.33 | **95.40** | 68.69 | 73.52 | 65.00 | 76.67 | 63.33 | 94.80 | 67.68 | 75.62 | 70.00 |
| | S | 76.67 | 53.33 | 94.80 | 66.83 | 72.91 | 65.00 | 80.00 | 76.67 | 95.80 | 67.82 | 80.07 | 78.34 |
| | D | 63.33 | 63.33 | 95.20 | 63.64 | 71.38 | 63.33 | 83.33 | 73.33 | **96.20** | 66.16 | 79.75 | 78.33 |
| S | N | 76.67 | 66.67 | 95.20 | 69.70 | 77.06 | 71.67 | 80.00 | 70.00 | **96.20** | 69.70 | 78.97 | 75.00 |
| | R | 83.33 | 60.00 | 94.20 | 68.69 | 76.55 | 71.66 | 83.33 | 73.33 | 95.00 | 70.20 | 80.47 | 78.33 |
| | S | 76.67 | 63.33 | 94.40 | 70.79 | 76.30 | 70.00 | **86.67** | 76.67 | 93.80 | 70.79 | **81.98** | **81.67** |
| | D | 80.00 | 60.00 | 94.60 | 69.19 | 75.95 | 70.00 | 83.33 | 66.67 | 94.40 | 68.18 | 78.14 | 75.00 |
| D | N | 73.33 | 63.33 | 95.00 | 70.20 | 75.47 | 68.33 | 76.67 | 73.33 | 95.80 | 68.69 | 78.62 | 75.00 |
| | R | 73.33 | 60.00 | 95.00 | 69.70 | 74.51 | 66.66 | 76.67 | 66.67 | 92.80 | 68.18 | 76.08 | 71.67 |
| | S | 80.00 | 56.67 | 93.80 | 66.83 | 74.33 | 68.34 | 80.00 | 73.33 | 93.60 | **71.78** | 79.68 | 76.66 |
| | D | 76.67 | 70.00 | 94.00 | 70.20 | 77.72 | **73.34** | 80.00 | 60.00 | 92.40 | 68.18 | 75.14 | 70.00 |

Table 8: **DeepSeek-R1-Distill-Qwen-32B:** "A24" is short for AIME24, and "A25" is short for AIME25. "Avg" is the average of four dataset results, and "Avg2" only counts for AIME24 and AIME25 results. We get the overall best performance with S-S under OT. Currently OT datasets show better results considering avg1 and on avg2. With the model becoming more capable, lower quality dataset with Input-Time Scaling can actually gain higher performance ceiling. While high quality datasets restrained more on Persona-N strategies.

| Train | Test | LIMO (High Quality) | | | | | | OT (Low Quality) | | | | | |
|---|---|---|---|---|---|---|---|---|---|---|---|---|---|
| | | A24 | A25 | MATH | GPQA | avg | avg2 | A24 | A25 | MATH | GPQA | avg | avg2 |
| N | N | 2.50 | 0.83 | 33.00 | 8.59 | 11.23 | 1.67 | 0.83 | 0.83 | 1.60 | 3.54 | 1.70 | 0.83 |
| | R | 3.33 | 0.83 | 28.80 | 9.60 | 10.64 | 2.08 | 1.67 | 0.83 | 20.60 | 4.04 | 6.78 | 1.25 |
| | S | 2.50 | 0.00 | 26.80 | **9.90** | 9.80 | 1.25 | 2.50 | 0.00 | 9.80 | 6.93 | 4.81 | 1.25 |
| | D | 0.83 | 1.67 | 23.00 | 6.06 | 7.89 | 1.25 | 1.67 | 3.33 | 11.20 | 2.53 | 4.68 | **2.50** |
| R | N | 0.83 | 1.67 | **37.00** | 9.09 | **12.15** | 1.25 | 1.67 | 0.00 | **39.60** | 9.09 | **12.59** | 0.83 |
| | R | 1.67 | 0.00 | 32.20 | 5.05 | 9.73 | 0.83 | 0.83 | 0.83 | 1.80 | 0.00 | 0.86 | 0.83 |
| | S | 1.67 | 0.83 | 28.40 | 6.93 | 9.46 | 1.25 | **3.33** | 0.83 | 1.60 | 0.00 | 1.44 | 2.08 |
| | D | 0.00 | 0.83 | 28.60 | 5.56 | 8.75 | 0.41 | 0.00 | 0.83 | 0.60 | 0.00 | 0.36 | 0.41 |
| S | N | 1.67 | 0.00 | 31.60 | 8.59 | 10.46 | 0.83 | 0.00 | 0.00 | 5.40 | 7.07 | 3.12 | 0.00 |
| | R | 3.33 | 0.00 | 32.20 | 8.08 | 10.90 | 1.67 | 1.67 | 0.00 | 0.80 | 0.51 | 0.74 | 0.83 |
| | S | 5.00 | 0.83 | 31.20 | 5.94 | 10.74 | **2.92** | 1.67 | 0.00 | 1.20 | 1.49 | 1.09 | 0.83 |
| | D | 0.83 | 0.83 | 25.80 | 5.56 | 8.26 | 0.83 | 0.83 | **4.17** | 1.60 | 1.01 | 1.90 | **2.50** |
| D | N | 2.50 | 2.50 | 34.00 | 6.06 | 11.27 | 2.50 | 1.67 | 0.00 | 35.40 | 8.08 | 11.29 | 0.83 |
| | R | 0.83 | 0.00 | 28.80 | 3.54 | 8.29 | 0.41 | 0.00 | 0.00 | 1.40 | 0.00 | 0.35 | 0.00 |
| | S | 3.33 | 0.83 | 28.80 | 2.97 | 8.98 | 2.08 | 0.83 | 0.00 | 1.20 | 0.99 | 0.76 | 0.41 |
| | D | 0.83 | 1.67 | 29.80 | 5.05 | 9.34 | 1.25 | 0.00 | 1.67 | 0.60 | 0.51 | 0.69 | 0.83 |

Table 9: **Llama-3.2-3B:** "A24" is short for AIME24, and "A25" is short for AIME25. "Avg" is the average of four dataset results, and "Avg2" only counts for AIME24 and AIME25 results. It shows a clear dominance of MATH learning abilities on LIMO than OT datasets.

| Train | Test | LIMO (High Quality) | | | | | | OT (Low Quality) | | | | | |
|---|---|---|---|---|---|---|---|---|---|---|---|---|---|
| | | A24 | A25 | MATH | GPQA | avg | avg2 | A24 | A25 | MATH | GPQA | avg | avg2 |
| N | N | 2.50 | 4.17 | 58.80 | 37.88 | 25.84 | 3.33 | 4.17 | 2.50 | 44.20 | 25.25 | 20.07 | 3.33 |
| | R | 9.17 | 1.67 | 55.40 | 40.40 | 26.66 | 5.42 | 5.83 | 5.00 | 43.00 | 23.74 | 19.18 | 5.42 |
| | S | 7.50 | 3.33 | 56.60 | 41.09 | 27.13 | 5.42 | **10.00** | 7.50 | 43.40 | 27.72 | 21.59 | **8.75** |
| | D | 8.33 | 7.50 | 50.80 | 38.38 | 26.25 | 7.92 | 5.00 | 5.00 | 40.00 | 21.72 | 19.60 | 5.00 |
| R | N | 3.33 | 1.67 | 59.40 | 37.88 | 25.57 | 2.50 | 4.17 | 1.67 | 48.20 | 26.26 | 20.07 | 2.92 |
| | R | 6.67 | 5.00 | 59.20 | 40.40 | 27.82 | 5.83 | 9.17 | 6.67 | 47.00 | 23.74 | **21.64** | 7.92 |
| | S | 6.67 | 4.17 | 59.20 | **45.54** | **28.89** | 5.42 | 6.67 | 5.00 | 46.20 | 28.22 | 21.52 | 5.83 |
| | D | 5.00 | 5.83 | 56.60 | 39.39 | 26.70 | 5.42 | 5.83 | 5.83 | 41.40 | 22.22 | 18.82 | 5.83 |
| S | N | 10.83 | 7.50 | 71.80 | 7.58 | 24.43 | 9.16 | 6.67 | 4.17 | 49.20 | 25.76 | 21.45 | 5.42 |
| | R | 11.67 | **10.83** | 71.00 | 5.05 | 24.64 | 11.25 | 7.50 | 3.33 | 49.60 | 22.73 | 20.79 | 5.42 |
| | S | 13.33 | 10.00 | **74.20** | 3.96 | 25.37 | 11.66 | 6.67 | 4.17 | 47.80 | 27.72 | 21.59 | 5.42 |
| | D | **15.00** | 9.17 | 72.00 | 6.06 | 25.56 | **12.09** | 6.67 | 7.50 | **52.80** | 19.19 | 21.54 | 7.08 |
| D | N | 9.17 | 4.17 | 59.20 | 40.91 | 28.36 | 6.67 | 7.50 | 3.33 | 44.20 | 25.25 | 20.07 | 5.42 |
| | R | 5.83 | 2.50 | 60.00 | 40.40 | 27.18 | 4.17 | 5.00 | 5.00 | 43.00 | 23.74 | 19.18 | 5.00 |
| | S | 8.33 | 4.17 | 59.40 | 39.60 | 27.88 | 6.25 | 6.67 | 5.00 | 43.40 | **31.19** | 21.57 | 5.83 |
| | D | 5.00 | 5.83 | 59.00 | 44.44 | 28.57 | 5.42 | 7.50 | **9.17** | 40.00 | 21.72 | 19.60 | 8.34 |

Table 10: **Llama-3.1-8B:** "A24" is short for AIME24, and "A25" is short for AIME25. "Avg" is the average of four dataset results, and "Avg2" only counts for AIME24 and AIME25 results. The difference on scores between LIMO and OT datasets is smaller than 3B models. With the models becoming more capable, the advantage of high-quality datasets is shrinking.

| Train | Test | LIMO (High Quality) | | | | | | OT (Low Quality) | | | | | |
|---|---|---|---|---|---|---|---|---|---|---|---|---|---|
| | | A24 | A25 | MATH | GPQA | avg | avg2 | A24 | A25 | MATH | GPQA | avg | avg2 |
| N | N | 5.83 | 2.50 | 44.60 | 8.59 | 15.38 | 4.17 | 0.83 | 2.50 | 1.00 | 1.52 | 1.46 | 1.67 |
| | R | 2.50 | 6.67 | 42.40 | 5.56 | 14.28 | 4.58 | 4.17 | 2.50 | 6.40 | 0.51 | 3.40 | 3.33 |
| | S | 5.83 | 5.00 | 42.40 | 5.94 | 14.79 | 5.42 | 2.50 | 2.50 | 3.80 | 1.49 | 2.57 | 2.50 |
| | D | 5.83 | 3.33 | 37.60 | 6.57 | 13.33 | 4.58 | 4.17 | 5.00 | 3.40 | 0.51 | 3.27 | 4.58 |
| R | N | 5.83 | 5.83 | **49.40** | 6.06 | 16.78 | 5.83 | 5.00 | 2.50 | 63.20 | 18.18 | 22.22 | 3.75 |
| | R | 4.17 | 5.83 | **49.40** | **9.60** | **17.25** | 5.00 | 4.17 | 7.50 | 2.40 | 1.52 | 3.90 | 5.83 |
| | S | 2.50 | 5.83 | 47.00 | 9.41 | 16.18 | 4.17 | 5.00 | 5.83 | 3.20 | 1.98 | 4.00 | 5.42 |
| | D | **7.50** | 6.67 | 44.40 | 6.57 | 16.29 | **7.08** | 5.00 | 5.83 | 2.80 | 1.01 | 3.66 | 5.42 |
| S | N | 3.33 | 3.33 | 46.40 | 9.09 | 15.54 | 3.33 | 5.83 | 5.83 | **65.20** | **24.24** | **25.28** | 5.83 |
| | R | 4.17 | 2.50 | 45.80 | 8.59 | 15.27 | 3.33 | 4.17 | 3.33 | 3.20 | 0.00 | 2.68 | 3.75 |
| | S | 5.00 | 6.67 | 43.40 | 7.43 | 15.62 | 5.83 | **6.67** | 2.50 | 2.00 | 0.00 | 2.79 | 4.58 |
| | D | **7.50** | 2.50 | 44.00 | 5.56 | 14.89 | 5.00 | 4.17 | **10.83** | 2.60 | 0.51 | 4.53 | **7.50** |
| D | N | 3.33 | **7.50** | 47.80 | 7.58 | 16.55 | 5.42 | 4.17 | 6.67 | 62.00 | 19.70 | 23.13 | 5.42 |
| | R | 5.00 | 6.67 | 47.00 | 4.04 | 15.68 | 5.83 | 5.00 | 2.50 | 3.60 | 0.51 | 2.90 | 3.75 |
| | S | 6.67 | 5.00 | 48.60 | 4.46 | 16.18 | 5.83 | **6.67** | 1.67 | 6.60 | 1.49 | 4.11 | 4.17 |
| | D | 5.00 | 6.67 | 47.20 | 8.08 | 16.74 | 5.83 | 2.50 | 5.00 | 5.60 | 0.51 | 3.40 | 3.75 |

Table 11: **qwen2.5-3B:** "A24" is short for AIME24, and "A25" is short for AIME25. "Avg" is the average of four dataset results, and "Avg2" only counts for AIME24 and AIME25 results. LIMO datasets show a clear advantage on MATH scores, which dominantes the avg.

| Train | Test | LIMO (High Quality) | | | | | | OT (Low Quality) | | | | | |
|---|---|---|---|---|---|---|---|---|---|---|---|---|---|
| | | A24 | A25 | MATH | GPQA | avg | avg2 | A24 | A25 | MATH | GPQA | avg | avg2 |
| N | N | 18.33 | 15.83 | 77.80 | 35.35 | 36.82 | 17.08 | 22.50 | 21.67 | 37.00 | 5.05 | 21.55 | 22.09 |
| | R | 18.33 | 14.17 | 74.20 | 27.78 | 33.62 | 16.25 | 17.50 | 21.67 | 36.80 | **8.59** | 21.14 | 19.59 |
| | S | 20.00 | 14.17 | 77.40 | **40.59** | 38.04 | 17.09 | 24.17 | 17.50 | 35.00 | 5.94 | 20.65 | 20.84 |
| | D | **22.50** | **24.17** | 74.20 | 37.37 | **39.56** | **23.34** | 21.67 | 21.67 | 27.00 | 6.06 | 19.10 | 21.67 |
| R | N | 18.33 | 16.67 | 77.40 | 32.83 | 36.31 | 17.50 | 24.17 | 17.50 | 43.00 | 2.53 | 21.80 | 20.84 |
| | R | 20.00 | 19.17 | 77.20 | 33.33 | 37.42 | 19.59 | 20.83 | 20.83 | 44.40 | 6.06 | 23.03 | 20.83 |
| | S | 20.83 | 17.50 | 77.20 | 29.70 | 36.31 | 19.16 | 19.17 | 21.67 | 44.00 | 3.47 | 22.08 | 20.42 |
| | D | 15.00 | 21.67 | 76.00 | 31.82 | 36.12 | 18.34 | 22.50 | **25.00** | 45.60 | 5.05 | 24.54 | 23.75 |
| S | N | 20.83 | 18.33 | 77.80 | 36.87 | 38.46 | 19.58 | 25.83 | 20.00 | 48.20 | 7.58 | 25.40 | 22.91 |
| | R | 18.33 | 17.50 | 76.40 | 30.30 | 35.63 | 17.91 | 25.00 | 19.17 | 45.40 | 6.06 | 23.91 | 22.09 |
| | S | 18.33 | 18.33 | 77.60 | 30.20 | 36.12 | 18.33 | 24.17 | 17.50 | 43.20 | 3.96 | 22.21 | 20.84 |
| | D | 15.83 | 18.33 | **79.00** | 31.82 | 36.24 | 17.08 | **26.67** | 22.50 | **48.60** | 6.57 | **26.09** | **24.59** |
| D | N | 20.00 | 17.50 | 77.00 | 37.37 | 37.97 | 18.75 | 21.67 | 19.17 | 44.60 | 3.54 | 22.25 | 20.42 |
| | R | 14.17 | 17.50 | 77.80 | 31.31 | 35.20 | 15.84 | 23.33 | 16.67 | 39.80 | 2.53 | 20.58 | 20.00 |
| | S | 21.67 | 16.67 | 78.00 | 32.67 | 37.25 | 19.17 | 24.17 | 23.33 | 44.40 | 3.96 | 23.96 | 23.75 |
| | D | 16.67 | 18.33 | 78.80 | 26.77 | 35.14 | 17.50 | 20.83 | 23.33 | 42.00 | 3.54 | 22.43 | 22.08 |

Table 12: **qwen2.5-7B:** "A24" is short for AIME24, and "A25" is short for AIME25. "Avg" is the average of four dataset results, and "Avg2" only counts for AIME24 and AIME25 results. LIMO datasets show a clear advantage on MATH scores,but the difference is becoming smaller. And OT shows a clear advantage on hard questions of AIME24 and AIME25.

| Train | Test | LIMO (High Quality) | | | | | | OT (Low Quality) | | | | | |
|---|---|---|---|---|---|---|---|---|---|---|---|---|---|
| | | A24 | A25 | MATH | GPQA | avg | avg2 | A24 | A25 | MATH | GPQA | avg | avg2 |
| N | N | 61.00 | 40.00 | 92.00 | 53.54 | 61.64 | 50.50 | 63.67 | 50.00 | 94.80 | 66.67 | 68.79 | 56.84 |
| | R | 43.33 | 40.00 | 90.20 | 51.01 | 56.14 | 41.67 | 60.00 | 53.33 | 91.20 | 66.67 | 67.80 | 56.67 |
| | S | **63.33** | 40.00 | 89.00 | 50.00 | 60.58 | 51.67 | 60.00 | 56.67 | 90.00 | 64.85 | 67.88 | 58.34 |
| | D | 40.00 | 36.67 | 87.20 | 51.52 | 53.85 | 38.34 | 60.00 | 60.00 | 89.40 | 61.62 | 67.76 | 60.00 |
| R | N | 59.33 | 36.67 | **93.80** | **68.69** | 64.62 | 48.00 | 40.00 | 26.67 | 92.80 | 68.18 | 56.91 | 33.34 |
| | R | **63.33** | 50.00 | 93.20 | 65.66 | 68.05 | 56.67 | 60.00 | 70.00 | 94.80 | 69.70 | 73.63 | 65.00 |
| | S | **63.33** | 53.33 | 93.40 | 66.34 | **69.10** | **58.33** | 73.33 | 53.33 | 94.00 | 67.33 | 72.00 | 63.33 |
| | D | 53.33 | **56.67** | 92.40 | 63.13 | 66.38 | 55.00 | 66.67 | 56.67 | 94.40 | 69.70 | 71.86 | 61.67 |
| S | N | 60.33 | 46.67 | 92.20 | 59.09 | 64.57 | 53.50 | 43.33 | 33.33 | 91.60 | 68.69 | 59.24 | 38.33 |
| | R | 43.33 | 43.33 | 90.20 | 58.08 | 58.74 | 43.33 | 73.33 | 60.00 | 94.40 | 71.72 | 74.86 | 66.67 |
| | S | 60.00 | 50.00 | 92.00 | 56.44 | 64.61 | 55.00 | 66.67 | 66.67 | 94.60 | 69.31 | 74.31 | 66.67 |
| | D | 53.33 | 50.00 | 89.40 | 59.09 | 62.96 | 51.67 | **76.67** | 70.00 | **95.00** | 66.67 | **77.09** | **73.34** |
| D | N | 55.67 | 43.33 | 92.20 | 58.59 | 62.45 | 49.50 | 33.33 | 36.67 | 91.80 | 66.16 | 56.99 | 35.00 |
| | R | 60.00 | 50.00 | 89.00 | 53.03 | 63.01 | 55.00 | 63.33 | 66.67 | 94.60 | **72.22** | 74.21 | 65.00 |
| | S | **63.33** | 46.67 | 90.60 | 57.43 | 64.51 | 55.00 | 70.00 | **73.33** | 94.20 | 67.82 | 76.34 | 71.67 |
| | D | 53.33 | 33.33 | 88.00 | 52.02 | 56.67 | 43.33 | 66.67 | 60.00 | 94.20 | 67.68 | 72.14 | 63.34 |

Table 13: **qwen2.5-32B:** "A24" is short for AIME24, and "A25" is short for AIME25. "Avg" is the average of four dataset results, and "Avg2" only counts for AIME24 and AIME25 results. Now OT dataset gain both Avg and Avg2 higher than LIMO dataset by a large margin. It improves more significantly on the hard problems when the model capabilities is enough.

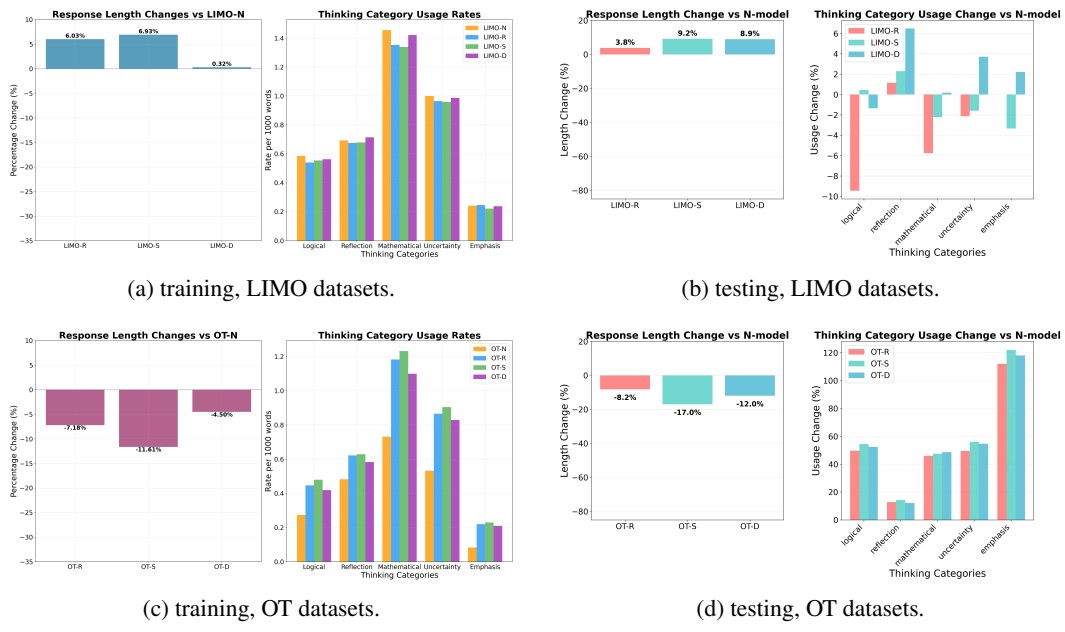

(a) training, LIMO datasets.

(b) testing, LIMO datasets.

(c) training, OT datasets.

(d) testing, OT datasets.

Figure 3: Performance comparison between training and testing strategies of DeepSeek-R1-Distill-qwen-32B: (a,c) is the training visualization; (b,d) is the testing visualization.

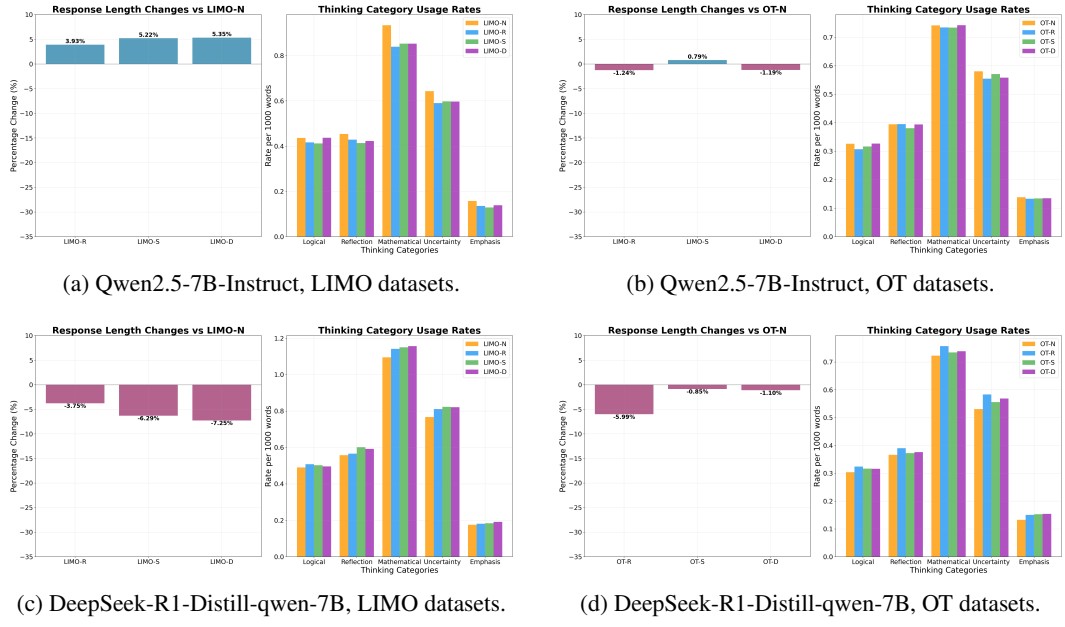

(a) Qwen2.5-7B-Instruct, LIMO datasets.

(b) Qwen2.5-7B-Instruct, OT datasets.

(c) DeepSeek-R1-Distill-qwen-7B, LIMO datasets.

(d) DeepSeek-R1-Distill-qwen-7B, OT datasets.

Figure 4: Performance comparison between training strategies

Figure 5: The distribution shift visualization of Qwen2.5-32B-Instruct on OT_1k dataset

