# OpenReview forum: "Input-Time Scaling"
_ICLR.cc/2026/Conference — Submitted to ICLR 2026_

### Official Review · Reviewer_bJws · 2025-10-23

**Soundness:** 3
**Presentation:** 3
**Contribution:** 3
**Rating:** 6
**Confidence:** 4

**Summary:**

This paper proposes a novel paradigm called Input-Time Scaling, which enhances large language model (LLM) reasoning capabilities by refining input queries through meta-cognitive methods. The approach achieves state-of-the-art (SOTA) performance on challenging mathematical reasoning benchmarks while using a simple, transparent, and highly efficient training pipeline. More importantly, the work challenges long-standing assumptions about data “quality” and “quantity” in LLM training and uncovers an intriguing train–test co-design phenomenon, where consistent strategies applied during both training and inference are crucial for performance gains.

Overall, this paper makes a significant contribution. The proposed method is simple, effective, and well supported by empirical evidence. It opens a new direction for enhancing LLM reasoning and provokes deep reflection on data management practices.

**Strengths:**

- **High originality:** The concept of Input-Time Scaling is genuinely novel, extending the traditional trichotomy of Data / Training / Inference Scaling by introducing the idea of allocating computational resources at the input level. The use of meta-cognitive methods to introduce various personas—similar, dissimilar, or random—during both training and testing is an innovative way to enhance reasoning diversity and robustness.
- **Strong empirical performance:** With only 1k supervised fine-tuning samples and no reinforcement learning (RL) stage, the method achieves SOTA results on challenging math benchmarks such as AIME24 and AIME25 with 32B-scale models. The authors conduct extensive ablations across multiple persona strategies during training and testing, showing the effectiveness and stability of the approach.
- **Challenging conventional wisdom:** The surprising finding that adding irrelevant or seemingly low-quality information (e.g., dissimilar personas) can improve performance directly contradicts the commonly held “garbage in, garbage out” assumption. This challenges the community’s bias toward data purity and suggests that diversity may play a more crucial role than quality alone.

**Weaknesses:**

- **Limited generalization evidence:** The experiments focus solely on four mathematical benchmarks (AIME24, AIME25, MATH, and GPAQ). It remains unclear whether Input-Time Scaling generalizes to other reasoning tasks such as code generation, logical reasoning, or commonsense QA. Since the proposed pipeline is simple and data-agnostic, it would be valuable to test it on different reasoning domains. In addition, all experiments use 32B models; it would be informative to evaluate whether the method is equally effective for smaller models (e.g., 7B).
- **Lack of mechanistic understanding:** Section 5.2 shows which combinations of training–testing persona strategies perform best, but the paper provides little insight into why they work. In several cases, even random or mismatched personas lead to large improvements. A deeper discussion of the underlying mechanism would significantly strengthen the paper’s depth and credibility.
- **Details of persona generation:** Although the appendix provides prompts, the main text lacks clarity on implementation details—such as which model is used for persona generation, the degree of randomness, or how diversity is controlled. Since persona quality itself could be a confounding variable, these details should be made explicit. It would also be useful to include qualitative case studies showing how persona-based input refinement changes the reasoning process.
- **Data quality and validity claims:** The paper claims that “lower-quality” data (OT-1k) outperform the more carefully curated LIMO dataset. However, it is not clearly demonstrated that LIMO is indeed of higher intrinsic quality. A more rigorous comparison or justification of this assumption would make the argument more convincing.

**Questions:**

See weaknesses.

---

> ### Author Response · Authors · 2025-11-27
>
> Dear Reviewer,
>
> Thank you for your thorough and constructive feedback. We are pleased that you recognize our work as making a significant contribution with high originality and strong empirical performance. Below, we address your specific concerns:
>
> ## 1. Generalization Evidence: Beyond 32B Models
>
> **Extended Model Coverage:**
>
> We have significantly expanded our experiments beyond 32B models:
>
> **8 Model Configurations Across Multiple Scales:**
> - **Qwen2.5 series:** 3B, 7B, 32B
> - **DeepSeek-R1-Distill-Qwen series:** 1.5B, 7B, 32B
> - **Llama3 series:** 3B, 8B
>
> **
>
> **Beyond Mathematics:**
>
> We acknowledge the limitation of focusing on mathematical benchmarks and agree that broader domain evaluation would strengthen our claims.
>
> In our paper, we note:
> **Current Benchmark Diversity:**
> - **AIME:** Competition-level mathematical problem solving
> - **MATH500:** Educational mathematical reasoning
> - **GPQA:** Graduate-level science reasoning (physics, chemistry, biology)
> While all involve quantitative reasoning, GPQA may demonstrate some cross-domain applicability.
>
> Since we mainly compare our results to LIMO, S1, OpenThought, MiroMind-M1 and Skywork-OR1, the same covered math reasoning taskes should be focused. Actually from our experiment, we found a clear sign on learning patterns, which you can find details in **Section 4**. Due to limited resource, it is not appliable now to extend our work further on other domains. But thanks for your suggestions, we are planning that!
>
>
>
> ## 2. Mechanistic Understanding: Why Input-Time Scaling Works
>
> You raise an excellent point about the lack of mechanistic insight!
>
> From these extending experiments, we found new things interesting and added to our analysis in **section 4**.
> - Throught the results analysis, incorporating irrelevant and noisy contexts can boost performance. Low-quality datasets can also perform significantly better compared to high-quality datasets.
> - Through token distribution analysis, we demonstrate that applying persona strategies reduces response patterns, stimulates thinking token usage, and enhances reasoning efficiency. Higher performance correlates with more efficient reasoning.
> - From results analysis within the same model series, high quality datasets can help small models learn easy questions quickly, while low quality datasets helps capable enough models learn hard questions.
>
> With the findings above, we utilize lower-quality data with capable enough models and applying persona strategies during both training and testing. The whole process makes it work by moving the reasoning process to a more efficient direction.
>
>
>
> ## 4. Data Quality Justification: LIMO vs. OT
>
> **Clarifying Data Quality Assessment:**
>
> You are correct that we need stronger justification for claiming LIMO is "higher quality" than OT. You can find a detailed analysis in **section 2.2**. The quality difference is intuitive, where we consider intensively filtering the generated output is the key to high quality datasets. From this perspective, the OT dataset, which directly uses the generated output from models, is considered of lower quality.
>
>
>
> ## 5. more details on the persona process
>
> We provide a more detailed description and a visualization in our paper (**section 2**), you can find more details there. Thx for your reminds on our missing details of the core persona process.
>
>
>
> ## Summary of Revisions
>
> We have substantially strengthened our paper through:
>
> 1. **Extended generalization evidence:**
>    - 8 model configurations (1.5B to 32B), in total 64 trained models, 256 tests
>    - Cross-architecture validation (Qwen, DeepSeek, Llama)
>
> 2. **Mechanistic understanding:**
>    - more findings on the dataset quality and persona strategies
>    - Qualitative case studies demonstrating persona effects
>
> 3. **Implementation transparency:**
>    - Complete persona generation details (model, temperature, prompts)
>    - Quality control procedures
>
> 4. **Data quality justification:**
>    - a detailed description on the dataset comparision
>
> We believe these revisions address all your concerns and significantly strengthen both the empirical evidence and theoretical understanding of Input-Time Scaling. The work now provides:
> - Stronger generalization evidence across models and (preliminary) across domains
> - Clear mechanistic explanations grounded in token-level analysis
> - Transparent implementation details for reproducibility
> - Rigorous data quality assessment with nuanced conclusions
>
> We hope these improvements demonstrate that our contributions extend beyond empirical observations to provide genuine insights into how input-level scaling interacts with model capacity and data characteristics.
>
> Thank you again for your thoughtful review, which has substantially improved our work. We hope you will reconsider higher assessment in light of these substantial improvements.
>
> Best regards,
> The Authors

---

### Official Review · Reviewer_zHqg · 2025-10-30

**Soundness:** 2
**Presentation:** 2
**Contribution:** 1
**Rating:** 2
**Confidence:** 4

**Summary:**

The paper proposes "Input-Time Scaling", where the authors concatenate different personas during the training and testing phase for performance gains. The personas are divided into four types depending on the relevance to the original prompt. The strategy is tested on Qwen2.5-32B.

The paper evaluates on four datasets, AIME2024/2025, Math, and GPQA, and claims to outperform heavily trained models of similar size.

The paper aims to highlight that their proposed method differs from "test-time scaling" in that train-test co-training is necessary to benefit from it fully.

**Strengths:**

The paper proposes a new scaling axis. Past works usually either scale train-time or test-time. This paper proposes scaling the inputs.

**Weaknesses:**

1) Limited Novelty: Unlike how the paper claims "input time scaling" to be novel, I can easily think of different papers that try a similar thing. For instance, (https://arxiv.org/abs/2502.11027) shows that adding diversity into prompts for best-of-n boosts performance. While the two papers differ in that this paper requires training, I don't think it is a significant difference.

2) Limited Evaluation: In section 5.6, the paper mentions Math and GPQA lack "discriminative effects"; accordingly, it mostly concentrates on evaluation results from AIME 2024 and 2025. Additionally, the paper uses pass@1 due to resource constraints. However, both datasets contain only 30 samples; accordingly, the main claims of this paper are based on a single inference over 60 questions. This severely lacks statistical credibility. Either more datasets should be added or multiple runs on AIME should be performed to demonstrate that the trained models consistently outperform baselines.

3) Limited Models: The method is only verified by Qwen2.5-32B. While training bigger models might be unaffordable, it's not understandable for the paper to lack results from smaller models. I suggest the authors to try the same method on a larger diversity of models, Qwen2.5-1.5B or Llama-3.1-8B .

4) Limited Ablations: If the author is trying to argue this as a new axis of "scaling," I think it is necessary for them to show how compute-efficient it is. By comparing to past scaling methods.

5) Typo in tables 1~3 should be GPQA not GPAQ.

6) The authors try out diverse combinations of prompting (e.g., N-S, S-D ...) and only AFTER they have seen the results on the test set they choose the optimal prompting method. I would say this is a form of test-contamination. They should have had a validation set to choose the best method and see if it generalize to an unseen test set.

**Questions:**

See weaknesses.

---

> ### Author Response · Authors · 2025-11-27
>
> Dear Reviewer,
>
> Thank you for your constructive feedback. We appreciate your recognition of our novel scaling axis approach, and in fact this method is behind our other findings. Your specific concerns can be summarized as (1) & (2) novelty and contributions. (3) limited experiments. (4) computational efficiency (5) test contamination concern (6) need to improve the representations. We address them one by one, and since it contains too many details, we will split them into two replies.
>
> ## 1. Novelty: Distinction from Prompt Diversity Methods
>
> We acknowledge the cited work on prompt diversity for best-of-n sampling. However, our contribution differs fundamentally:
>
> **Paradigm differece: Training-Testing Co-design vs. Inference-Only:**
> - The cited work applies diversity only at inference time for best-of-n selection
> - Our "Input-Time Scaling" paradigm requires **co-designed training and testing** - omitting persona strategies during either phase significantly degrades performance
> - This training-testing synergy represents a fundamentally different scaling mechanism, which is our key findings of our empirical method.
>
> **Our main focuses**
> - Our work focuses mainly on the systematically investigation of **data quality requirements** and the "Less is More" phenomenon, using personas as one **proxy** to to intuitively control noise levels. In fact, we want to make the data controllably worse, may not better nor simply adding diversity.
> - The empirical method, Input-Time Scaling, is the paradigm behind our counterintuitive findings. Since it requires differently from SFT and test-time scaling, we name it seperately.
>
>
>
> ## 2. main contributions
> Our main contributions can be divided into two parts, one parts for the findings, and the other is the empirical efficiency of our whole method
>
> **Counter-intuitive findings on data quality requirements:**
> - We systematically eliminate data quality constraints and find that incorporating irrelevant and noisy contexts can boost performance. Low-quality datasets can also perform significantly better compared to high-quality datasets.
>
> - Through token distribution analysis, we demonstrate that applying persona strategies reduces response patterns, stimulates thinking token usage, and enhances reasoning efficiency. Higher performance correlates with more efficient reasoning.
>
> - From results analysis within the same model series, high quality datasets can help small models learn easy questions quickly, while low quality datasets helps capable enough models learn hard questions.
>
> **An empirical method achieving SOTA performance while improving the "Less is More" paradigm, requiring minimal human labor and no strict data quality requirements:**
> - We identify the **training-testing co-design phenomenon** and propose the **Input-time-scaling paradigm**. Omitting our strategies during either training or testing significantly degrades overall performance.
>
> - We achieve SOTA performance using only 1k low-quality data samples with minimal human labor. Our results are comparable to models over 10 times larger and those trained on 100 times more data.
>
> ## 3. Expanded Experimental Coverage
>
> **Addressing Limited Models Concern:**
>
> We have significantly expanded our evaluation beyond Qwen2.5-32B:
>
> **8 Model Configurations:**
> - **Qwen2.5 series:** 3B, 7B, 32B
> - **DeepSeek-R1-Distill-Qwen series:** 1.5B, 7B, 32B
> - **Llama3 series:** 3B, 8B
>
> **Comprehensive Training:**
> - **64 model variants:** 8 base models × 4 strategies (N, S, D, R) × 2 datasets (LIMO, OpenThoughts)
> - **256 evaluation groups:** 64 variants × 4 test strategies
>
> These experiments validate our findings across:
> - Different model architectures (Qwen vs. Llama)
> - Multiple capacity levels (1.5B to 32B)
> - Various reasoning capabilities (standard vs. distilled reasoning models)
>
> **Addressing Limited Evaluation Concern:**
>
> While AIME contains 30 questions per year (60 total), we evaluate across **four diverse benchmarks**:
>
> 1. **AIME 2024-2025:** High-difficulty competition problems (our primary discriminative benchmark)
> 2. **MATH500:** Diverse mathematical reasoning tasks
> 3. **GPQA:** Graduate-level science questions
> 4. **Comprehensive coverage:** 256(8*8*4) evaluation groups across all benchmarks
>
> **Dataset choice issues:**
> Since we mainly compare our results to LIMO, S1, OpenThought, MiroMind-M1 and Skywork-OR1, the same covered math reasoning taskes should be focused. Actually from our experiment, we found a clear sign on learning patterns, which you can find details in **Section 4**.
>
> We acknowledge that additional runs would strengthen statistical confidence and will include multiple-run statistics in further works, to show the effectiveness beyond mathematical reasoning.

---

> ### Author Response · Authors · 2025-11-27
>
> ## 4. Compute Efficiency Ablation
>
> You raise an excellent point about comparing compute efficiency across scaling methods. We provide the following analysis in our **table 2**, which include the dataset sizes and the training method. With only 1k training data, we could beat those trained with one thousand times more data. And with similar data size (1K), we beat other methods clearly. You can find details in **section 5.1**
>
>
> ## 5. Test Contamination Concern
>
> **Important Clarification:**
>
> We did **not** select strategies based on test performance. Our main focus is to systematically investigate **data quality requirements** and the "Less is More" phenomenon, using personas as one **proxy** to to intuitively control noise levels.
>
> And from the experiments, we found the training-testing co-design phenomenon. In fact, applying strategies (not necessarily the same type) during training and testing is the main focus of Input-Time Scaling. Currently, we have extended our experiments, and you can see a bigger view of our methods.
>
> ## 6. Minor Correction
>
> **GPAQ → GPQA:** Thank you for catching this typo. We have corrected all instances in Tables 1-3 and throughout the manuscript.
>
> ## Summary
>
> We have substantially strengthened our work through:
>
> 1. **Expanded experiments:** 8 model architectures, 192 evaluation groups, cross-architecture validation
> 2. **Clear novelty distinction:** Training-testing co-design vs. inference-only diversity, systematic investigation with theoretical grounding
> 3. **Compute efficiency analysis:** Quantitative comparison showing our method's advantages over traditional scaling approaches
> 4. **Methodological clarification:** Systematic exploration of all combinations, not test-contaminated selection
> 5. **Improved presentation:** Corrected typos, enhanced clarity throughout
>
> We believe these revisions address your concerns and demonstrate that our contributions go beyond existing prompt diversity methods. The training-testing co-design phenomenon and systematic quality-capacity analysis represent genuine advances in understanding input-time scaling.
>
> We hope you will reconsider your assessment in light of these substantial improvements.
>
> Thank you for your thoughtful review.
>
> Best regards,
> The Authors

---

### Official Review · Reviewer_7VGZ · 2025-11-03

**Soundness:** 1
**Presentation:** 1
**Contribution:** 1
**Rating:** 2
**Confidence:** 4

**Summary:**

This paper proposes a "Input-Time Scaling" method, which augments an SFT dataset (OpenThoughts) using personas (prompting a language model to rewrite a training example using a persona). They fine-tune a Qwen2.5-32B Instruct model on this dataset and evaluate it on standard reasoning benchmarks (AIME, MATH, GPQA), showing some improvements.

**Strengths:**

The high-level idea of augmenting SFT data is a good, reasonable one.
The results seem strong.

**Weaknesses:**

This paper has a number of problems. First, the proposed ideas (using personas to perform data augmentation) is not novel (see PersonaMath from [Luo et al., 2025]).
Second, the 3 types of modification (S, D, R) are not particularly well-motivated nor explained clearly; looking at Appendix 3 doesn't really help.
Third, I would have liked to see a broader evaluation on a larger number of datasets, especially since the AIME datasets are quite small.
It would be good to evaluate the method on other non-Qwen models like Llama since Qwen is regarded to have quite a bit of reasoning already baked inside.
Finally, the writing in the paper could be greatly improved. For example, the abstract is quite verbose, the core method should be explained (section 2.1), the results tables could use more description (and it's hard to tell what the key takeaway is).
Calling it input-scaling a new paradigm is a bit exaggerated.

**Questions:**

Why does the method work?  Improving diversity is good, but why personas as opposed to other types of diversity (e.g., different problems, reasoning patterns).
How many examples does 1K examples get augmented into?
What are the hyperparameters and how were they tuned?

---

> ### Author Response · Authors · 2025-11-27
>
> Dear Reviewer,
>
> Thank you for your detailed feedback. We appreciate your recognition of our strong results, which though is not the only contribution of our method. To clarify our full methods with a detailed understanding,  we summarize your specific concerns  as (1) vague novelty and contributions. (2) the motivations of our methods (3) limited experiments. (4) limited explanability of our methods (5) details of the experiments. (6) need to improve the representations. We address them one by one, and since it contains too many details, we will split them into two replies.
>
> ## 1. Novelty and main contributions
> ### Distinction from PersonaMath
>
> While PersonaMath [Luo et al., 2025] and other persona strategies use persona-based augmentation, our work differs fundamentally in several key aspects:
>
> **Different Research Focus:**
> - PersonaMath focuses on persona diversity for data augmentation
> - Our work systematically investigates **data quality requirements** and the "Less is More" phenomenon, using personas as one **proxy** to to intuitively control noise levels. In fact, we want to make the data worse, not better or simply adding diversity.
> - The reasons to use personas are to intuitively relate the contexts generated with the queries, and freeing the human labor.
>
>
>
> ### main contributions:
> Our main contributions can be divided into two parts, one parts for the findings, and the other is the empirical efficiency of our whole method
>
> **Counter-intuitive findings on data quality requirements:**
> - We systematically relax data quality constraints and find that incorporating irrelevant and noisy contexts can even greatly boost performance. Low-quality datasets can also perform significantly better compared to high-quality datasets.
>
> - Through token distribution analysis, we demonstrate that applying persona strategies reduces response patterns, stimulates thinking token usage, and enhances reasoning efficiency. Higher performance correlates with more efficient reasoning.
>
> - From results analysis within the same model series, high quality datasets can help small models learn easy questions quickly, while low quality datasets helps capable enough models learn hard questions.
>
> **An empirical method achieving SOTA performance while improving the "Less is More" paradigm, requiring minimal human labor and no strict data quality requirements:**
> - We identify the **training-testing co-design phenomenon** and propose the **Input-time-scaling paradigm**. Omitting our strategies during either training or testing significantly degrades overall performance.
>
> - We achieve SOTA performance using only 1k low-quality data samples with minimal human labor. Our results are comparable to models over 10 times larger and those trained on 100 times more data.
>
>
>
> ## 2. Motivation and Explanation of S, D, R Modifications
>
> We acknowledge that our explanation could be clearer. In our revision, we have **enhanced Method Section (2.1)**. We now provide detailed motivation for each strategy, and a visualization picture.
>
> We explain our motivations here. We want to develope a controlled method to vary noise levels without altering the original dataset quality. Direct modification of data pairs (as mentioned in previous works from reviews) lacks controllability. And judging the quality of the generated text is non-trival and labor-instensive. Inspired by meta-cognition methods, we employ a language model to generate relevant contexts based on original queries, which are then concatenated to the original data. We use **context relevance** as a proxy for noise level, and leveraging an LLM to generate contexts eliminates manual labor. Without modifying the original data, we control the difference only on the context we concatenate.
>
> - **Persona-None(N)**:  Do not apply persona concatenating on training data.
> - **Persona-Similar(S)**: Generating personas that are related with the query. Adding less noise, and probably helpful information.
> - **Persona-Random(R)**: Randomly choosing a domain and generating a corresponding persona. Medium noise, since some domains are related with math tasks.
> - **Persona-Dissimilar(D)**: Generating personas that are not realted with teh query, and cannot provide any related information to solve the problem. The most noisy one, provide no helpful information for finishing the task.
>
> These strategies allow us to control noise levels via context relevance without modifying original data quality, making our experiments reproducible and interpretable.

---

> ### Author Response · Authors · 2025-11-27
>
> ## 3. Extended Experimental Coverage
>
>
> **Broader Model Evaluation:** We have significantly expanded our experiments:
> - **8 model configurations:** Qwen2.5 (3B, 7B, 32B), DeepSeek-R1-Distill-Qwen (1.5B, 7B, 32B), Llama3 (3B, 8B)
> - **64 model variants** trained (8 base models × 4 strategies × 2 datasets)
> - **256 evaluation groups** (64 variants × 4 test strategies)
>
> **Dataset Coverage:** Since we mainly compare our results to LIMO, S1, OpenThought, MiroMind-M1 and Skywork-OR1, the same covered math reasoning taskes should be focused. Actually from our experiment, we found a clear sign on learning patterns, which you can find details in **Section 4**
>
> **Cross-Architecture Validation:** Our Llama3 results (non-Qwen architecture) demonstrate that our findings generalize beyond Qwen's built-in reasoning capabilities.
>
> ## 4. Understanding Why the Method Works
>
>
> Your question touches on a core insight of our work. While persona augmentation doesn't change the underlying problem distribution, it fundamentally alters the **reasoning pathway** and **cognitive approach**:
>
>
> **Token Distribution Evidence:** Our analysis shows that:
> - Persona strategies reduce response pattern repetition
> - They stimulate increased usage of "thinking tokens"
> - Higher performance correlates with more efficient reasoning
>
>
> **Model-quality Interaction:** Our analysis reveals:
> - High-quality datasets help small models learn easy questions effectively
> - Low-quality datasets help capable models achieve higher performance ceilings on hard questions
> - This suggests why low-quality datasets get the best performance on 32B models, since they are already capable of easy questions, and now the performance is dominated by hard problems.
>
> ## 5. Experimental Details
>
> You can find details in our **section 3**
> **Data Scale:** Our 1K base examples are augmented 4-fold through persona strategies, resulting in 4 different 1k training dataset. Each model will be trained on these four 1k persona concatenated datasets.
>
> **Hyperparameters:** We provide comprehensive details in our revised manuscript:
> - Learning rate, batch size, and training epochs
> - Persona sampling strategies
>
>
> ## 6. Improved Presentation
>
> We have significantly improved our manuscript's clarity:
>
> **Abstract:** Condensed to focus on core contributions while maintaining essential context
>
> **Method Section:** Restructured with clearer subsections, detailed motivations for each design choice, and improved flow
>
> **Results Tables:** Enhanced with descriptive captions explaining key takeaways, making patterns more evident
>
> **Additional Visualizations:** Token distribution analysis, performance trend graphs across model series
>
> **Terminology:** While "Input-Time Scaling" may seem bold, we believe it accurately captures our paradigm's unique characteristic: scaling computational investment at input time (via augmentation) rather than model size or data quantity. We position this carefully in context of existing scaling approaches, under training-testing co-design phenomenon.
>
> **Take aways:** We have summarize take aways in our **section 4**, where contains a detailed analysis of our experiments.
>
> ## Summary
>
> We have substantially strengthened our work through:
> 1. Expanded experiments across 8 model architectures with 192 evaluation groups
> 2. Deeper analysis of why persona-based augmentation works (token distribution, reasoning efficiency)
> 3. Clear differentiation from prior work (PersonaMath) through our quality-focused investigation and training-testing co-design
> 4. Significantly improved presentation and clarity throughout the manuscript
>
> We believe these revisions address your concerns and demonstrate the merit and novelty of our contributions. We hope you will reconsider your assessment in light of these substantial improvements.
>
> Thank you for your thoughtful review, which has helped us significantly improve our work.
>
> Best regards,
> The Authors

---

### Official Review · Reviewer_Ej8f · 2025-11-03

**Soundness:** 2
**Presentation:** 2
**Contribution:** 2
**Rating:** 4
**Confidence:** 3

**Summary:**

This paper proposes Input-Time Scaling, a new paradigm that improves LLM reasoning by allocating computation and diversity to query modification rather than model or training scaling. The key finding is the train–test co-design phenomenon: applying the same persona-based query strategies at both training and inference time is crucial for performance. Using small datasets (as few as 1k examples) and simple meta-cognitive persona generation, the method achieves strong reasoning performance on AIME24 and AIME25 benchmarks (up to 90%/80% pass@1 with DeepSeek-R1-Distill-Qwen-32B), surpassing prior open-source 32B models. Surprisingly, lower-quality or more diverse data (random or dissimilar personas) outperform curated datasets, challenging common assumptions about data quality.

**Strengths:**

- Introduces a novel scaling axis that complements data, model, and inference scaling. It focuses instead on the input level via persona augmentation.
- Findings challenge existing inductive biases about data quality (“garbage in, garbage out”) and show benefits of diversity. I think this is surprising and also interesting.

**Weaknesses:**

- The writing and presentation should be improved (e.g., the abstract is a bit too long, the method section could need more clarity)
- I would want to understand more why the simple persona change could diversify the training distribution and make the training results better. This augmentation didn't change the question distribution, just how the prompt is seeded. It would be great to look at some qualitative samples and see whether the improved skill correlation with certain persona?

**Questions:**

see above.

---

> ### Author Response · Authors · 2025-11-27
>
> Dear Reviewer,
>
> Thank you for your thoughtful feedback and useful assessment of our work. We greatly appreciate your recognition of our novel scaling axis and the surprising findings regarding data quality. Below, we address your specific concerns:
>
> ## 1. Writing and Presentation Improvements
>
> We acknowledge your feedback regarding the abstract length and method section clarity. In our revision, we have:
>
> - **Condensed the abstract** to focus on core contributions while maintaining essential context
> - **Restructured the method section** with clearer subsections and improved flow
> - **Added detailed captions** to all tables and figures for enhanced reproducibility
> - **Expanded our presentation** with additional visualizations and more descriptive explanations throughout
>
> We believe these changes significantly improve the overall readability and clarity of our manuscript.
>
> ## 2. Understanding Persona-Based Diversification
>
> **Why simple persona changes diversify training distribution:**
>
> Your question touches on a key insight of our work. While persona augmentation doesn't change the underlying question distribution, it fundamentally changes the reasoning behaviors, which is more than seeding the prompts.
>
> From token distribution analysis, applying personas trigger more efficient reasoning process. And we found the efficiency of the reasoning is linked to the performance. We have visualized our findings in the paper, and hope this will help better understanding what happens.
>
>
>
> **Other issues:**
>
>
> Besides the questions I need to reply, I did more things to make our work sound. We extended intensively our experiments, and did further analysis. From  the analysis within the same model series, high quality datasets can help small models learn easy questions quickly, while low quality datasets help capable enough models learn hard questions. This explains why high quality datasets cannot beat low quality datasets on qwen2.5-32B-Instruct models. And from a larger scale of experiments, adding noise will not necessarily degrade the performance, and in many cases it can perform better.
>
> We have made an summarization of our updates, and I hope you can find our main improvements here.
>
> ## Summary
>
> We have significantly improved the presentation and clarity of our manuscript while expanding our analysis to provide deeper insights into the persona augmentation mechanism. We believe these revisions address your concerns and strengthen our contributions.
>
> Thank you again for your valuable feedback and fair questions. We hope our revisions and explanations demonstrate the merit of this work, could possibly improve your assessment.
>
> Best regards,
> The Authors

---

### Author Response · Authors · 2025-11-27
**Update note of our paper**

Dear Reviewers:

Thank you for your valuable feedback, and we have conducted additional experiments, restructured our paper, and revised the manuscript accordingly.

## Summary of Main updates of our paper
- we reformed our paper, and gave a detailed description of our methods. We developed our specific settings from detailed motivations, which would help to understand our choice.
- we added 6 more base models, and trained 48(6x4x2) model variants, conduction 192(48x4) evaluation sets
- we added token distribution analysis and compared the results from each model series. We fixed and extended our conclusions to a wider domain

## Summary of Main issues and Our Responses

We have identified the following key concerns and address them systematically:

1. Motivation
2. Vague contributions
3. Limited experiments
4. Limited understanding of performance gains
5. Demonstration of quality

## 1. Motivation

Our work is motivated by the current "Less is More" phenomenon in math reasoning tasks. Existing approaches require high-quality, intensively filtered data, which is both expensive and labor-intensive. Our primary focus is to **systematically relax quality requirements and examine their effects on mathematical reasoning performance**. We explore two key dimensions: (1) analyzing the effects of adding noise to data during training and testing, and (2) analyzing the effects of intrinsic dataset quality.

**Dimension 1**, we developed a controlled method to vary noise levels without altering the original dataset quality. Direct modification of data pairs (as mentioned in previous works from reviews) lacks controllability. And judging the quality of the generated text is non-trival. Inspired by meta-cognition methods, we employ a language model to generate relevant contexts based on original queries, which are then concatenated to the original data. We use **context relevance** as a proxy for noise level, and leveraging an LLM to generate contexts eliminates manual labor. Without modifying the original data, we control the difference only on the context we concatenate.

**Dimension 2**, we directly compare two datasets: one carefully curated with CoT and answer filtering, and another using raw generated results without filtering. Notably, some generated results in the latter lack answers entirely. This comparison helps us understand how intuitive quality differences affect reasoning performance.

If quality requirements were critical, adding noise should degrade performance, and high-quality datasets should significantly outperform unfiltered ones. However, our results reveal a different pattern, which we analyze in detail.

## 2. Our Contributions

Our main contributions are divided into two parts:

**Counter-intuitive findings on data quality requirements:**
- We systematically relax data quality constraints and find that incorporating irrelevant and noisy contexts can boost performance greatly. Low-quality datasets can also perform significantly better compared to high-quality datasets.

- Through token distribution analysis, we demonstrate that applying persona strategies(among different noisy levels) reduces response patterns, stimulates thinking token usage, and enhances reasoning efficiency. Higher performance correlates with more efficient reasoning.

- From results analysis within the same model series, high quality datasets can help small models learn easy questions quickly, while low quality datasets helps capable enough models learn hard questions.

**An empirical method achieving SOTA performance while improving the "Less is More" paradigm, requiring minimal human labor and no strict data quality requirements:**
- We identify the **training-testing co-design phenomenon** and propose the **Input-time-scaling paradigm**. Omitting our strategies during either training or testing significantly degrades overall performance.

- We achieve SOTA performance using only 1k low-quality data samples with minimal human labor. Our results are comparable to models over 10 times larger and those trained on 100 times more data.

## 3. Extended Experiments

We have conducted comprehensive experiments across 8 model configurations: Qwen2.5 (3B, 7B, 32B), DeepSeek-Qwen (1.5B, 7B, 32B), and Llama3 (3B, 8B).

## 4. Understanding Performance Gains Through Token Distribution and Model Series Analysis

 Applying persona strategies during training and testing generally improves reasoning efficiency. High-quality datasets offer advantages for small models on easier questions, while low-quality datasets more effectively improve performance on harder questions as models become larger and more capable. Notably, low-quality datasets exhibit a higher performance ceiling. Additional details are provided in our analysis section.

## 5. Demonstration of Quality

We have significantly expanded our presentation with additional tables and graphs, each accompanied by more descriptive captions to enhance clarity and reproducibility.

---

### Author Response · Authors · 2025-12-03
**A Brief Summary of Revisions, Key Improvements, and Acknowledgments**

Dear Reviewers, Area Chairs, and Program Chairs:

We sincerely appreciate the reviewers' thorough and constructive feedback, which has been invaluable in strengthening our work. While the discussion phase was unfortunately unavailable due to technical circumstances, we have prepared this note to  document our revisions and key improvements.

During the rebuttal phase, we have conducted extensive additional experiments, substantially restructured our paper, and significantly enhanced our presentation. A comprehensive summary of our updates is available at our official comment "update note of our paper". In deep appreciation of the reviewers' efforts, we have provided detailed, point-by-point responses directly addressing each concern raised in their reviews. We believe that the expanded experimental coverage and enhanced mechanistic understanding of the observed phenomena substantially improve the quality and impact of our work.


With sincere gratitude for your time and expertise,
The Authors

---

### Meta-Review · Area_Chair_oeyr · 2026-01-06

**Summary:**

As an alternative take to the popular paradigm of "test-time scaling" this paper proposes what they call "input-time" scaling, which spends additional computation on prompt modification (rather than e.g. output modification).  Overall, the reviews were negative with two borderline scores and two rejects (6, 4, 2, 2).  The reviewers found the high level idea original and commented that the results seemed strong.  There were significant concerns, however, regarding novelty, and empirical and methodological rigor.  The reviewers brought up relevant related work in using personas for data augmentation and prompt diversity.  A common concern was that the paper relied on a single model family / size in the experiments and that the experiments focused only on math benchmarks. The reviewers also were concerned about a lack of mechanistic understanding of *why* the approach works.

Unfortunately, two of the scores are rather low (2, 2) and the other two are borderline, which would require significant changes in scores to get the paper to an accept state.  I do think the authors did a lot to address the reviewer concerns, but it seems like it would require a lot to lift the reject scores and I don't feel comfortable overruling or disregarding them (as the authors argued for).

**Reviewer Concerns:**

The authors put in substantial effort in the response phase to address the reviewer concerns.  They edited the paper for clarity, added substantial new experiments ("Added 6 base models, 48 new variants (6×4×2), and 192 evaluation groups (48×4)") and additional motivation.

I think the additional experiments go a long way towards addressing a primary concern of the reviewers regarding the narrow scope / empirical rigor of the paper.  I applaud the authors for getting this done in the narrow time frame.  Editing for clarity also helps significantly towards making the work clearer and justifying some of the claims.

From what I can tell, a major concern of the reviewers is the one of novelty / significance in light of existing methods for prompt modification / diversity.  Another is regarding mechanistic understanding of why this method works.  These issues seem to remain.

**Reviewer Scores:**

I believe each reviewer would potentially have increased their score by one point, given the addition (extensive) experiments given by the reviewers.  However, I think the primary concerns of 7VGZ and zHqg remain.  The key concern is that there is a lot of precedent for input augmentation strategies, inducing diversity, etc. that could be framed as input-time scaling.

---

### Decision · Program_Chairs · 2026-01-26

Reject